# Rehabilitation Technologies by Integrating Exoskeletons, Aquatic Therapy, and Quantum Computing for Enhanced Patient Outcomes

**DOI:** 10.3390/s24237765

**Published:** 2024-12-04

**Authors:** Fabio Salgado-Gomes-Sagaz, Vanessa Zorrilla-Muñoz, Nicolas Garcia-Aracil

**Affiliations:** 1Systems Engineering and Automation Department, University Miguel Hernández of Elche, 03202 Elche, Spain; fabio.salgado@goumh.umh.es; 2Bioengineering Institute, University Miguel Hernández of Elche, 03202 Elche, Spain; nicolas.garcia@umh.es; 3Institute on Gender Studies, University Carlos III of Madrid, Getafe, 28903 Madrid, Spain

**Keywords:** exoskeletons, aquatic therapy, proportional–integral–derivative, fuzzy logic controllers, quantum computing

## Abstract

Recent advancements in patient rehabilitation integrate both traditional and modern techniques to enhance treatment efficacy and accessibility. Hydrotherapy, leveraging water’s physical properties, is crucial for reducing joint stress, alleviating pain, and improving circulation. The rehabilitation of upper limbs benefits from technologies like virtual reality and robotics which, when combined with hydrotherapy, can accelerate recovery. Exoskeletons, which support and enhance movement, have shown promise for patients with neurological conditions or injuries. This study focused on implementing and comparing proportional–integral–derivative (PID) and fuzzy logic controllers (FLCs) in a lower limb exoskeleton. Initial PID control tests revealed instability, leading to a switch to a PI controller for better stability and the development of a fuzzy control system. A hybrid strategy was then applied, using FLC for smooth initial movements and PID for precise tracking, with optimized weighting to improve performance. The combination of PID and fuzzy controllers, with tailored weighting (70% for moderate angles and 100% for extensive movements), enhanced the exoskeleton’s stability and precision. This study also explored quantum computing techniques, such as the quantum approximate optimization algorithm (QAOA) and the quantum Fourier transform (QFT), to optimize controller tuning and improve real-time control, highlighting the potential of these advanced tools in refining rehabilitation devices.

## 1. Introduction

The rehabilitation of patients with or without neuropathies has been increasingly studied in recent decades, with a growing focus on improving both the effectiveness and accessibility of treatments. New therapies are focused on more specific and individualized approaches, targeting the unique needs of each patient to enhance recovery outcomes. These therapies often incorporate advanced technologies and tailored exercise regimens to address the particular challenges presented by different types of neuropathies and other conditions. In the case of lower and upper limb rehabilitation, new and emerging therapies present a significant challenge due to the complexity and diversity of movements these limbs perform. Upper limbs, including arms and hands, comprise a wide range of daily activities, from basic tasks like dressing and eating to more complex functions like writing or manipulating small objects. Traditional therapies for these limbs are now often complemented by advanced technologies, such as virtual reality and robotics, which offer new ways to stimulate and recover functionality [1]. The combination of physical exercises with visual and tactile stimuli in a controlled environment can accelerate recovery and improve therapeutic outcomes [2], which is achieved using rehabilitation exoskeleton systems tailored to patients’ needs and conditions.

Actually, the rehabilitation of patients with or without neuropathies has been the subject of increasing study in recent decades, focusing on improving both the effectiveness and accessibility of treatments. New therapies are oriented towards more specific and individualized approaches, addressing the unique needs of each patient to enhance recovery outcomes.

### 1.1. Exoskeletons in Rehabilitation: Applications, Design Challenges, and Technological Advancement

The integration of exoskeleton technology into rehabilitation represents a significant evolution in therapeutic approaches for patients with motor impairments. This section explores how exoskeletons function as both assistive devices and therapeutic tools, offering critical support for individuals recovering from injuries, strokes, and other neurological conditions. By understanding the mechanics and applications of these devices, their role in personalized rehabilitation and the challenges that arise from designing them to accommodate the complexities of human movement can be appreciated.

Rehabilitation exoskeletons are applied to aid in the recovery of people who have suffered accidents, strokes, or other illnesses [3]. Exoskeletons, which are external structures that support and enhance the body’s movement, have shown great potential in rehabilitation in lower and upper limbs. These devices can be programmed to assist in a variety of movements, from walking to more complex activities, depending on the patient’s specific needs. They can assist patients in recovering their motor skills, providing support and assistance in movements that are weakened or paralyzed. Additionally, exoskeletons can adapt and adjust to the changing needs of the patient as they progress in their rehabilitation, offering continuous and personalized support.

The development of robotic rehabilitation technologies, such as exoskeletons, requires incrementally addressing the challenges associated with the complexity of limb movements. Limbs, especially in a rehabilitation context, involve multiple degrees of freedom, which entails the simultaneous control of various joints and body segments. This high level of biomechanical and technical complexity demands a gradual design approach, starting with the implementation of a single degree of freedom to minimize risks and ensure effective basic functionality before advancing to more complex solutions.

A single degree of freedom is an effective strategy in the early stages of exoskeleton development. For example, research conducted by Dey et al. [4] demonstrates that focusing on a single degree of freedom can enhance performance in aquatic environments, where buoyancy reduces joint load and improves movement fluidity. Similarly, this approach has proven valuable in the rehabilitation of stroke patients. Previous studies indicate that nursing care robots designed with a single degree of freedom can assist stroke patients in performing bicep physiotherapy exercises, enabling movements that may be challenging to accomplish independently. This design not only optimizes the functionality of the robot but also prioritizes patient safety and comfort during the rehabilitation process [5]. Additionally, implementing a single degree of freedom allows the robot to adapt more effectively to the individual capabilities of each patient, thereby enhancing the overall efficacy of the treatment and promoting better recovery outcomes. There are also studies and a production of exoskeletons with one degree of freedom, such as the Stride Management Assist system developed by Honda, the orthosis developed at Yonsei University, or the single-degree-of-freedom (DoF) wearable robot designed at the University of Teknologi. For the control of exoskeletons, fuzzy and neuro-fuzzy systems have been studied in exoskeletons [6].

Research has also highlighted that employing a single degree of freedom aids in validating control algorithms and improving human–machine interaction. These methodologies facilitate adjustments and enhancements in control systems before incorporating additional degrees of freedom. This incremental approach is essential for achieving stable and efficient control, especially in systems where sensory feedback and user adaptability are critical for therapeutic success.

Finally, the Assisted Rehabilitation and Measurement (ARM) guide employs a single degree of freedom, which enhances its efficacy as a rehabilitation tool for upper limb functions. This design simplifies the rehabilitation process, allowing patients to engage in targeted reaching exercises without the complexity of managing multiple joints. By focusing on linear movement, the ARM guide not only reduces costs, making it an economical option for rehabilitation centers, but also enhances safety and comfort during therapy sessions. This approach aligns with evidence supporting the effectiveness of single-degree-of-freedom systems in promoting motor recovery, as they enable precise control and better adaptability to individual patient needs [7,8].

Robotic rehabilitation technology is at the forefront of modern rehabilitation practices, revolutionizing how healthcare professionals approach the recovery process. This section highlights key advancements in robotic systems, specifically focusing on various exoskeletons developed for both lower and upper limb rehabilitation. By examining the impact of these technologies on mobility, independence, and emotional well-being, we can better understand their transformative potential in enhancing patient care.

Exoskeletons represent a revolution in medical assistance and rehabilitation, providing a powerful tool to improve the mobility and quality of life for people with various medical and physical conditions. One exoskeleton example is the EksoGT exoskeleton, which is designed to assist people with lower limb paralysis or weakness in walking. This exoskeleton has been used in clinical settings to help patients with stroke, spinal cord injuries, and other neuromuscular conditions perform walking movements that promote rehabilitation and recovery. Studies have shown that the EksoGT not only enhances patients’ mobility and independence but can also have a positive impact on their emotional and psychological well-being, providing a greater sense of autonomy and improvement in quality of life. By enabling individuals to perform movements that were previously challenging or impossible, the EksoGT helps bridge the gap between rehabilitation and daily life activities, allowing patients to engage more fully in their personal and social environments. This increased engagement can lead to improved mental health outcomes and greater satisfaction with life [9].

In the case of upper limb rehabilitation, the ReFlex exoskeleton aims to help patients recover fine motor skills and dexterity, needed for daily activities like eating and dressing. The device uses soft robotics and wearable technology to assist and guide the user’s arm movements, providing targeted support for motor recovery [10]. For the lower extremities, including the hips, knees, and ankles, the exoskeleton systems are not very different. For example, the ReWalk exoskeleton enables individuals with spinal cord injuries to stand, walk, and even climb stairs. This device has been pivotal in patient´s rehabilitation, allowing them to perform walking exercises in a controlled and safe environment, and has shown significant improvements in muscle strength and motor function [11]. Other exoskeletons provide rehabilitation in a wide range of daily activities: for example, HAL (Hybrid Assistive Limb) covers a multiactivity proposal because it is designed for both upper and lower limb rehabilitation. HAL supports the wearer’s movements through bioelectrical signals, providing assistance based on the user’s intention. This exoskeleton has shown promise in improving mobility and strength, particularly for patients with severe motor impairments [12].

### 1.2. Integrating Hydrotherapy and Robotics

Combining hydrotherapy with robotic rehabilitation offers a novel approach to patient recovery, harnessing the therapeutic properties of water alongside the precision of robotic assistance. This section discusses the synergy between hydrotherapy and robotic systems, illustrating how this integration enhances therapeutic outcomes. By exploring the benefits of aquatic environments in rehabilitation, we gain insights into how these methods can address specific challenges faced by patients with varying conditions, from spasticity to musculoskeletal issues.

Rehabilitation techniques with exoskeletons systems have evolved significantly, ranging from traditional methods to advanced technological innovations. Among these techniques, the use of water therapy has been utilized recently and for long-term recovery conditions or diseases [13]. Hydrotherapy, or aquatic therapy, leverages the physical properties of water, such as buoyancy, resistance, and hydrostatic pressure, to offer an ideal therapeutic environment. These properties allow patients to perform exercises more safely and effectively, minimizing the risk of injury and stress on the joints. The use of hydrotherapy is effective for patient recovery [14] for various reasons, one of which is pain [15,16] and spasticity reduction [17]. Spasticity is a condition where muscles are excessively tight and rigid, leading to involuntary movements and difficulty controlling the muscles [18]. This muscle stiffness can be painful and can limit the range of motion, significantly affecting the patient’s functional capacity and participation. Hydrotherapy can alleviate these symptoms by providing an environment where water buoyancy reduces the strain on muscles and joints, allowing for smoother and less painful movements. Additionally, the water helps relax tense muscles and facilitates exercises that would otherwise be difficult or impossible on land. The reduction in spasticity and relief from pain contribute to greater comfort during exercise, enabling more effective and less stressful rehabilitation for patients.

Moreover, water allows for low-impact exercises, which is beneficial for the joints [19] and can alleviate chronic musculoskeletal pain [20]. Buoyancy reduces the apparent body weight, facilitating movements that would be difficult or impossible on land. The application of hydrotherapy creates a synergistic effect that enhances both physical and psychological rehabilitation outcomes [21]. The water’s resistance strengthens muscles and improves cardiovascular capacity without the need for additional weights, while the exoskeleton adjusts the level of assistance or resistance according to the user’s progress. This integrated approach not only enhances physical strength and endurance but also optimizes the recovery of key motor functions. The constant sensory feedback from the water improves proprioception and balance, providing dual neuromuscular and sensory stimulation that accelerates physical recovery and reinforces the mind–body connection, essential for long-term rehabilitation success.

For all the reasons indicated, hydrotherapy rehabilitation has been demonstrated to be useful for treatments in patients who have suffered strokes [13,22,23]. Especially, the aquatic environment also acts as a natural cushion, reducing sudden or uncontrolled movements, which is especially beneficial for patients with spasticity [24]. Also, patients with arthritis, patients after anterior cruciate ligament (ACL) reconstruction [25,26], and patients with back pain [27] could obtain benefits with treatment based on hydrotherapy. These benefits range from improved strength and flexibility to the recovery of mobility and balance. In the case of stroke patients, hydrotherapy can play a crucial role in enhancing coordination and motor function, aiding in the recovery of essential daily living skills. In the case of patients affected by arthritis, the hydrotherapy helps to reduce pain and inflammation in the joints while promoting a wider and less painful range of motion [28,29,30]. For patients who have undergone ACL reconstruction, hydrotherapy facilitates a more effective recovery by allowing exercises that strengthen muscles without placing excessive strain on the newly operated joint reconstructions [25,26]. Additionally, for those with back pain, the aquatic environment offers significant relief by reducing pressure on the spine and surrounding muscles, allowing for greater participation in therapeutic activities [31,32,33,34] and contributing to an overall improvement in quality of life.

The integration of hydrotherapy with robotic-assisted movements can enhance motor learning and facilitate a more comprehensive rehabilitation process [14]. So, the combination of an exoskeleton with an aquatic environment for rehabilitation offers significant benefits that enhance the quality of treatment and patient recovery.

Aquatic exoskeletons promise to revolutionize rehabilitation by integrating the supportive and therapeutic effects of water with the precision and adaptability of robotic exoskeletons [35]. Underwater robotic covers leverage water buoyancy to reduce impact on the body [36] and the exoskeleton’s ability to offer targeted assistance, and these systems enable patients to perform movements and exercises with greater ease and safety [37]. In recent years, studies on exoskeletons for rehabilitation purposes have been increasing. Some studies have already incorporated the benefits of aquatic therapies into these exoskeletons [or even exoskeletons like SUBAR, which has a control algorithm inspired by aquatic therapy [38]. This enables movements and exercises that would be difficult or impossible on land. The exoskeleton provides additional support by guiding and assisting the user’s movements, ensuring they are performed safely and in a controlled manner. This combination reduces the risk of injury and allows for a more gradual and manageable rehabilitation process. This synergy not only enhances the efficiency of the rehabilitation process but also accelerates recovery, improves functional outcomes, and boosts patient motivation and engagement. As a result, hybrid aquatic exoskeletons represent a significant advancement in personalized, effective, and accessible rehabilitation solutions.

### 1.3. Role of Data Analytics in Aquatic Rehabilitation

In the era of big data and advanced computing, integrating data analytics into rehabilitation practices is crucial for optimizing patient outcomes. This section delves into the role of programmable logic controllers (PLCs) and their ability to automate and refine rehabilitation processes. By examining how data collection and real-time monitoring contribute to personalized therapy, we can appreciate the importance of technology in enhancing the efficiency and effectiveness of rehabilitation interventions.

In addition to the inherent benefits of the hybrid aquatic exoskeleton system, integrating precise control strategies with automated advanced tools such as PLCs (programmable logic controllers) could help to optimize the rehabilitation processes. Integrating PLCs into hybrid aquatic exoskeleton systems enhances rehabilitation by providing precise control over various movements, including stretches and range of motion. PLCs not only provide the automatization of rehabilitation processes but also offer an accurate and repeatable control, which allows healthcare professionals to design and monitor treatment protocols more effectively [39]. They enable real-time data collection and remote monitoring, which facilitates ongoing supervision of patient progress without requiring constant physical presence from the therapist. This capability ensures that treatments can be finely tuned to the individual needs of each patient, enhancing personalization and precision in therapy.

The advanced control offered by PLC systems also includes significant safety features. These systems can be configured with multiple levels of protection to minimize the risk of injury, contributing to a safer treatment environment. Although the initial investment in PLC technology can be higher, the long-term benefits—such as reduced errors, improved efficiency, and increased reliability—often result in substantial cost savings and better management of healthcare resources. PLC systems improve the integration of data in exoskeletons and aquatic therapy setups, allowing for more adaptive and responsive rehabilitation. For example, a PLC-controlled exoskeleton can adjust the support or speed of movements according to the user’s strength and endurance. This adaptability not only enhances the effectiveness of treatments but also reduces the risk of injury, promoting a safer and more efficient recovery process [40].

Moreover, PLCs provide the integration of control systems, such as the PI (proportional–integral controller) and PID (proportional integral–derivative controller). The PI and PID offer additional advantages in terms of treatment precision and adaptability. The PI controller, which utilizes proportional and integral components, is effective in maintaining stability and correcting cumulative errors in repetitive movements, resulting in smooth and consistent trajectory tracking. On the other hand, the PID controller, which includes a derivative component in addition to the proportional and integral parts, allows for quicker reactions to sudden or unexpected changes in system behavior [41], such as alterations in speed or in the resistance offered by water. For this reason, a hybrid PID and fuzzy logic controller (FLC) approach can be particularly beneficial, combining the robust response capabilities of the PID controller with the adaptive [42], rule-based decision-making of fuzzy logic [43,44]. This hybrid control strategy enhances the system’s ability to handle complex, nonlinear dynamics and uncertainties, leading to improved overall performance and user experience in aquatic rehabilitation scenarios. This hybrid control approach ensures that the exoskeleton can adapt in real time to variations in rehabilitation conditions, providing the ideal balance between smoothness and responsiveness. In an aquatic environment, where movement dynamics can be less predictable due to the buoyancy and resistance of water, this advanced control guarantees that exercises are performed precisely and safely, enhancing the efficiency of the rehabilitation process and ensuring patient safety.

However, traditional control systems such as PIDs and FLCs often struggle to handle the complexity and nonlinearity inherent in real-time applications, particularly in dynamic systems like exoskeletons [44,45]. PID controllers, while effective in many scenarios, can struggle with nonlinearities and varying system dynamics that are characteristic of exoskeletons. They often require constant tuning and adjustment to maintain optimal performance as the system conditions change. This tuning process, especially for highly dynamic systems, can become cumbersome and may not be sufficient for handling rapid or unpredictable variations in system behavior. Similarly, FLCs, although more flexible and capable of handling uncertainty, still face challenges when dealing with complex interactions and the continuous evolution of the system’s state in real time.

Integrating quantum computing into these control systems offers a promising solution to overcome these challenges. Quantum computing has the potential to process and analyze vast amounts of data simultaneously, making it well suited to handle the high levels of complexity and nonlinearity present in dynamic systems [46] like exoskeletons. By leveraging quantum algorithms, control systems can optimize their performance in real time, even in the presence of noise and rapidly changing conditions. This capability would allow for more precise adjustments and faster responses to unexpected variations in the system’s behavior, leading to smoother and more efficient operation.

### 1.4. New Directions in Aquatic Rehabilitation Robotics

As we look towards the future, emerging technologies such as quantum computing hold the potential to further transform rehabilitation practices. This section explores the prospects of integrating quantum computing with existing rehabilitation technologies, focusing on the advantages it may bring in handling complexity and improving control systems. By envisioning the future landscape of rehabilitation technology, we can anticipate advancements that may significantly enhance patient recovery and redefine therapeutic methodologies.

The integration of quantum computing into rehabilitation technology promises to address the inherent complexities of dynamic systems like exoskeletons. Traditional control systems often struggle with nonlinearities and the variability of patient movements, which can hinder effective rehabilitation. Similar to the challenges faced in optimizing enterprise management systems, where classical algorithms encounter performance issues [47], rehabilitation technologies can benefit from quantum algorithms to enhance their adaptive capabilities. Quantum computing’s ability to process vast amounts of data and perform calculations at unprecedented speeds can offer more responsive control mechanisms. This enhanced capability allows for real-time adjustments to treatment protocols, ensuring that the support provided by robotic systems is precisely tailored to each patient’s evolving needs. Moreover, the application of quantum Monte Carlo methods, as explored in optimizing automated systems, could further improve the effectiveness of rehabilitation technologies, combining quantum and classical approaches to deliver tailored patient care.

Additionally, quantum computing can enhance the robustness of these control systems against noise, a common issue in real-world applications. Quantum error correction techniques and algorithms designed to work in noisy environments can ensure that the control signals remain accurate and reliable, even under challenging conditions. This integration could reduce the need for constant manual tuning and adaptation, allowing the control system to automatically adjust its parameters to maintain optimal performance. As a result, the combination of quantum computing with traditional control methods could revolutionize the way dynamic systems like exoskeletons are managed, offering more advanced, adaptive, and efficient solutions for real-time control and rehabilitation processes.

Moreover, practical examples have emerged where quantum computing principles are being tested in robotic applications. According to recent research, quantum computing is poised to significantly impact robotics, enabling the development of more powerful and intelligent robots that leverage quantum cloud services and co-processors. For instance, a position paper discusses the potential applications of quantum computing in various aspects of robotics, including artificial intelligence, machine learning, sensing, and kinematics, suggesting that quantum algorithms could revolutionize robotic perception and navigation [48]. Additionally, studies exploring quantum-like (QL) models for robot perception indicate that incorporating concepts like state superposition and probabilistic interference can enhance the capabilities of robots with limited sensing abilities [49]. These advancements illustrate how quantum computing can refine robotic systems’ interaction with complex environments, improving navigation and obstacle avoidance and ultimately leading to more sophisticated, personalized, and efficient rehabilitation technologies that enhance patient outcomes.

### 1.5. Implementation Process: Transition from Initial Prototyping to Advanced Patient Care

Before integrating patients into robotic rehabilitation systems, a robust testing phase focuses on technological refinement to ensure the results of the aquatic system integrated process. In this **pre-clinical phase**, exoskeleton prototypes are developed with a **single degree of freedom**, often starting with basic joint movements like knee articulation. Hydrotherapy is simulated within these systems to measure how the **properties of water—such as buoyancy and resistance—impact the exoskeleton’s movement**. The inclusion of **quantum computing in later stages** introduces enhanced adaptability, enabling the **management of multiple degrees of freedom** and more intricate motor functions, which will eventually support complex movements in patients. This careful, phased approach ensures the safety, precision, and efficiency of the integrated system, paving the way for its application in clinical settings. This comprises two differentiates phases (see Figure 1 with the representation).

In the initial phase of developing the implementation process for the aquatic rehabilitation robot, a focus is proposed based on the evaluation and development of the rehabilitation system before involving patients. This process begins with an initial assessment of the system, where potential use cases for exoskeletons and hydrotherapy are identified. This first step establishes a clear framework for technology development, ensuring that the objectives are aligned with clinical needs. Next, the prototype testing of the exoskeleton is conducted, facilitating the construction of a functional model with a single degree of freedom.

This phase also allows for observation of how the device behaves in simulated environments, providing valuable insights into its movement control. Integration of hydrotherapy simulations is also performed at this stage, examining how water properties affect the performance of the exoskeleton without human intervention. The data collection and analysis integration step is a key step where data on the exoskeleton’s performance are gathered and mechanical responses are refined. In this regard, the implementation of PIDs and FLCs enables real-time adjustments, ensuring that the system responds appropriately to variations in the simulated environment.

At the end of this phase, it is crucial to design the quantum system that will be used in the next stage. This involves selecting the most relevant variables for the functioning of the quantum system, ensuring that these variables optimize the control and adaptability of the exoskeleton. This selection of variables is based on data collected during the testing phase and is essential for ensuring that the quantum system can be effectively integrated into the rehabilitation process.

A second phase would focus on the integration of patients and the application of quantum computing to develop a more advanced system that manages multiple degrees of freedom. This initiation relies on the advanced development of the exoskeleton, utilizing quantum computing to enhance the precision and adaptability of the device’s control. This allows the system to handle complex movements, which is fundamental for meeting the diverse needs of patients. Integration of hydrotherapy for patient use is carried out in parallel, where hydrotherapy sessions are combined with the advanced exoskeleton. This approach leverages the properties of water to reduce the physical effort of the patient during rehabilitation.

Subsequently, the advanced data analysis phase comes into play, using machine learning techniques to adjust therapy sessions in real time based on the individual needs of the patient. Adjustments of PIDs and FLCs are also enhanced with quantum computing, allowing the system to respond more effectively to patient signals. In this phase, patient feedback is collected, which is essential for adapting sessions and improving the rehabilitation experience. Finally, the evaluation of the final outcomes is conducted to measure improvements in mobility, emotional well-being, and independence of the patients. This evaluation not only determines the success of the treatment but also provides valuable insights for future rehabilitation strategies.

## 2. Materials and Methods

The aim of this article is to evaluate the combination of PIDs and FLCs in a lower limb exoskeleton designed for rehabilitation therapy. This study seeks to compare the effectiveness of each control strategy, both individually and in combination, by analyzing their capabilities in managing various aspects of system control, including movement precision, adaptability to dynamic changes, and the overall effectiveness in an aquatic rehabilitation setting. This approach ensures a comprehensive evaluation of how these controllers can optimize the performance of rehabilitation technologies in clinical applications applied for phase 1 of “Evaluation and Development of the system” before tests with patients.

The presented work is part of the main NOHA project, an innovative robotic-aquatic rehabilitation system designed to facilitate the recovery of patients with severe neurological disorders, such as spinal cord injuries, strokes, or multiple sclerosis. NOHA combines the advantages of robotics and aquatic therapy, offering a system that provides personalized and adaptive rehabilitation, especially for patients who cannot benefit from conventional robotic rehabilitation due to issues like spasticity, pain, or limited range of motion. This system leverages the properties of water to reduce joint load and increase rehabilitation effectiveness. The NOHA system is designed to adapt to the specific needs of each patient, allowing adjustments in the degrees of freedom based on the type and severity of the injury. Its modularity enables the integration of different components, such as the hydraulic system and hybrid actuators, to provide precise and effective rehabilitation. This modular approach also facilitates the customization of treatment, allowing the system to be adjusted to various mobility and strength requirements necessary for optimal rehabilitation. The flexibility in design is essential to address the complexity of neurological disorders and ensure that the system can provide effective support at different stages of the recovery process. Figure 2’s upper portion of the illustration relates to the application of the upper and lower limb aquatic robotic rehabilitation system, while the lower part depicts the sequence of activation of micro-water jets of the hybrid actuators. It assists the movements of the shoulder and elbow inside the tank.

This modular system consists of two main assemblies: the hydraulic system that powers the actuator and the hybrid actuator itself. One of the main advantages of this device is that it uses water flow to generate most of the torque needed for the desired movement, with fine adjustments being made possible by the integration of a high-precision electric motor. Furthermore, immersion in water offers additional benefits, such as weightlessness, joint unloading, and resistance to movement, allowing control over water properties like density, hydrostatic pressure, viscosity, temperature, and buoyancy. Both assemblies, the hydraulic system and the actuator, are essential for the proper functioning of the product, although they can be sold separately to, for example, power two different hybrid actuators with a single hydraulic system. This would be the case of a robotic device with two coupled degrees of freedom, placed in a single water tank and powered by the same hydraulic system.

As for the hybrid actuator, it consists of an electric motor, its driver, links, and microjets. The microjets can be positioned closer to or further from the center of rotation, depending on the customer’s needs, to modify the generated torque. Likewise, the number of jets can be adjusted to increase the torque, making the system adaptable to different types of rehabilitation.

To bridge the gap between simulation-based results and future clinical applications, the development of the NOHA rehabilitation system also involves the careful design and testing of control mechanisms that can handle the complexities of human movement in an aquatic environment. The combination of control strategies like PIDs and FLCs is particularly relevant in this context, as the system must be able to adapt to the varying dynamics of water while maintaining precise control over movement. Simulations and laboratory testing have been crucial in refining these control systems before transitioning to real-world use. By integrating the PID, known for its precision, with the FLC, which excels in managing nonlinear conditions, the NOHA system aims to address the challenges posed by water’s variable resistance and buoyancy, offering a more adaptive and stable rehabilitation experience.

As the NOHA system continues to evolve, the research focuses on optimizing these control strategies to ensure they are effective in real-world clinical settings. This modular robotic system, designed for both upper and lower limb rehabilitation, benefits from the combined use of water’s natural properties and advanced robotic technologies. At this stage of the research, the results presented are derived from simulations and controlled laboratory testing of the prototype rather than real-world implementation. The current prototype is still in the design and development phase and has not yet been tested with human subjects. Consequently, the results reflect the performance of the control system under simulated conditions rather than actual clinical use. Detailed descriptions of the implementation process and real-world testing will be provided as the prototype advances to clinical evaluation, which will require adherence to the European Union Medical Device Regulation (MDR) and other relevant standards to ensure safety and efficacy. This approach allows us to refine the prototype and its control mechanisms before proceeding to more rigorous human trials.

The control of the NOHA exoskeleton has been approached through an advanced integration of the PID and FLC to address the challenges associated with their complex dynamics. The PID controller is renowned for its effectiveness in systems with linear characteristics, providing precise adjustments and stabilization. This controller is particularly useful for maintaining a consistent position and minimizing deviations, which is used in applications where accuracy is paramount. However, PID controllers can struggle with highly nonlinear systems and varying disturbances, necessitating frequent recalibration to sustain optimal performance. On the other hand, FLCs excel in managing nonlinearities and accommodating complex, imprecise inputs, making them ideal for systems with uncertain or variable conditions. Despite their strengths, FLCs can face issues such as delayed stabilization and less predictable responses.

The initial analysis involved implementing control using a PID controller on the PLC. During the first attempts to stabilize the system with 10 trials, it was observed that the derivative (D) term of the controller caused the system to be more unstable and jittery. This behavior was due to the derivative term reacting quickly to changes in the rate of error, which amplified oscillations rather than mitigating them. Due to this instability, we switched to a PI (proportional–integral) controller, which excludes the derivative term and focuses only on the proportional (P) and integral (I) terms.

The proportional (P) controller adjusts the output based on the current error, while the integral (I) term corrects for accumulated past errors to eliminate residual error. This simplified approach helped reduce oscillations and improved stability, but the system still required adjustments to optimize its performance. After 30 adjustment attempts, the iterative process was justified by the need to thoroughly assess how each variation in parameters affected the system’s behavior. Each attempt provided valuable insights into the system’s dynamics and allowed for fine-tuning of the controller parameters to improve stability and performance. The number of attempts reflects a systematic approach for exploring different configurations, identifying and correcting configuration issues and refining parameters for a more predictable and efficient response. Despite these efforts, it was determined that the system’s behavior still did not meet the desired requirements, leading to the need for a new control strategy.

In response, a fuzzy control system was developed within the PLC, incorporating all necessary functions: a fuzzifier, membership functions, rules, and a defuzzifier. Fuzzy control is particularly useful in systems where relationships between variables are nonlinear or difficult to model accurately using traditional methods. This system design was carried out following the guidelines established by the IEC 61131-7 standard [50], which provides a standardized framework for implementing fuzzy control in PLC systems.

For the output function, the center of gravity method for simple and discrete elements, singletons (CoGS), was used, which is also supported by the IEC 61131-7 standard. This method provides a precise numerical approximation for membership function terms that have specific and defined numerical values, unlike a curve that represents a range of values. The calculation of this method is performed using (1), which is used to obtain a centralized value representing the output of the fuzzy system. This approach ensures greater precision in decision making and contributes to system stability, better aligning with the nonlinear characteristics and inherent uncertainties in the control process.
(1)$$\frac{\sum_{i=1}^{p}\left[U_{i}*a_{i}\left(t_{k}\right)\right]}{\sum_{i=1}^{p}\left[a_{i}\left(t_{k}\right)\right]}$$
where U_i_ is the support point at position i and a_i_ is the value of the singleton at position U_i_ with i ranging from 1 to the number of singleton elements.

For the created defuzzifier block, the above formula is:(2)$$Output=\frac{\sum_{i=1}^{n}\left[{MDX}_{i}*X_{i}\right]}{\sum_{i=1}^{n}\left[{MDX}_{i}\right]}$$
where MDX_i_ is the response of the rules for X_i_ support point, with i ranging from 1 to the number of singleton elements.

Around 30 controller adjustment tests were conducted due to the system’s complexity and the need to achieve optimal performance. The sensitive nature of control systems, especially those with complex or nonlinear dynamics, requires multiple adjustments to thoroughly explore how each parameter change affects the system’s behavior. Each test provides data that allow for the fine-tuning of the controller parameters and balancing of factors such as stability, precision, and response. This iterative process helps identify and correct errors, validate the controller’s performance under different conditions, and adapt the system to real-world variations. By conducting these extensive tests, it ensures that the final controller configuration is robust and reliable, minimizing the likelihood of errors and guaranteeing consistent and effective performance.

To leverage the strengths of both PIDs and FLCs, a hybrid control strategy was developed. This approach utilizes the FLC during the initial phase of movement to ensure a smooth and gradual initiation, which is crucial for applications like patient rehabilitation where abrupt movements can be detrimental. The fuzzy controller’s ability to handle nonlinearities and adapt to varying conditions makes it well suited for this phase. Once the movement begins, control gradually transitions to the PID controller to ensure precise tracking and stability during the final part of the movement. This combination effectively harnesses the smooth, adaptive nature of fuzzy logic and the precise, stabilizing features of PID control, leading to enhanced overall performance and reliability of the exoskeleton system.

The implementation of this hybrid approach involved refining the weighting between the PID and FLC controllers. Initial tests with a simple linear weighting scheme demonstrated improvements in both smoothness and stability of the exoskeleton’s movements. To further optimize the control performance, a more sophisticated weighting function was employed, based on a cosine function. This advanced function allowed for a dynamic adjustment of the control emphasis between PIDs and FLCs depending on the phase of the movement. The result was a more nuanced and effective control system that could handle various movement angles and scenarios with greater precision. This refinement led to improved performance compared with using either control method in isolation.

The following diagram (see Figure 3) shows the process of transitioning from the PID method to achieving the hybrid implementation of PID and FLC methods in the aquatic exoskeleton.

Specific challenges such as “noise” during the return phase of movement were addressed by isolating the PID controller for this phase. This adjustment reduced unwanted disturbances and improved the overall quality of the return movement. Robustness testing confirmed that the hybrid control system remains effective across a range of angles close to those originally studied, proving its versatility and adaptability. This comprehensive approach, integrating PID and fuzzy control with advanced weighting techniques, not only enhances the smoothness and stability of movements but also ensures that the exoskeleton performs reliably in diverse conditions.

The results from all combinations are proposed for programming in a quantum computing environment, such as quantum approximate optimization algorithm (QAOA) and variational quantum eigensolver (VQE), which allows for advanced parameter optimization in complex and dynamic systems. QAOA is used to address **combinatorial optimization problems by** approximating the best possible solution through quantum evolution in multiple steps, leveraging both quantum and classical computation to optimize system parameters [51]. On the other hand, VQE is employed to find the **minimum eigenvalues** of matrices associated with physical systems, making it particularly useful for **quantum simulations** [52]. VQE combines quantum computing with classical optimization methods to approximate ground-state energy values in molecular systems [53].

This proposal includes the integration of these quantum algorithms to address real-time optimization problems and improve the precision and adaptability of the exoskeleton’s control system. Quantum computing, with its capability to process and analyze large volumes of data at superior speeds, offers an innovative approach to overcoming the limitations of traditional control methods like PID and FLC algorithms.

The article also explores how quantum programming can enable more efficient adaptation to changing environmental and user conditions, providing a more robust and flexible solution for managing the dynamic variables of the system. This can lead to significant improvements in system stability and real-time response to variations, resulting in more effective and personalized rehabilitation for exoskeleton users.

Finally, a comparative analysis between traditional methods and quantum approaches is presented, highlighting the potential advantages of quantum computing in controlling complex dynamic systems and its impact on the future development of advanced rehabilitation technologies.

## 3. Results

This section is divided into the following subsections: Control Techniques for Exoskeletons; Prototyping and Testing; and Controller Performance and Optimization.

### 3.1. Control Techniques for Exoskeletons

In the realm of exoskeleton control, traditional methods such as PID (proportional–integral–derivative) controllers remain prominent. These controllers can be optimized using techniques like Ziegler–Nichols [41,54] or Cohen–Coon [54] or through trial-and-error approaches. For example, in the case of predictable and repetitive tasks, PID controllers are designed for single-input single-output (SISO) systems, making them effective for simpler control scenarios [55]. However, their limitations become apparent when managing systems with substantial nonlinearities. In slightly nonlinear systems, PID controllers require frequent adjustments to maintain performance, and in highly nonlinear systems, their performance can degrade significantly, requiring the use of alternative control strategies [56].

In contrast, FLCs offer advantages in managing multiple input variables and handling nonlinear systems more effectively [57]. These controllers employ a set of rules and membership functions to process inputs in a manner that resembles human reasoning, making them particularly useful for complex and adaptive control tasks. Moreover, FLCs are especially beneficial in dynamic scenarios, such as aquatic environments, where conditions could change quickly [58]. For this reason, the introduction of adaptive neuro-fuzzy systems further enhances this capability, allowing the controller to adjust and learn from its environment dynamically. Recent research has highlighted the benefits of integrating neural networks with FLCs, leading to more sophisticated adaptive control solutions. Despite these advancements, PID controllers are still relevant, especially when combined with modern techniques like optimal and robust control to address system uncertainties, disturbances, friction, and unmodeled dynamics.

### 3.2. Prototyping and Testing

The development of a new exoskeleton prototype is part of a broader project that aims to refine and expand its capabilities. This initiative is crucial for advancing rehabilitation technologies, particularly in aquatic environments where water properties can significantly improve recovery outcomes. The prototype focuses on the lower limb section and is designed to operate effectively in a water-based rehabilitation environment. This design consideration takes into account the buoyancy and resistance provided by water, which can reduce patient strain while still allowing for controlled movement. The aim is to automate specific movements programmed by a physiotherapist, allowing treatment sessions to continue without their constant presence, thus increasing the efficiency and accessibility of therapy. For testing purposes, an aquarium was selected to simulate the water-based rehabilitation environment, as illustrated in Figure 4. We started by analyzing an exoskeleton for the lower limb (Figure 4 left), and we modeled a part of this exoskeleton (Figure 4 center); the proposed model was built and attached to the aquarium to carry out the tests (Figure 4 right). The prototype was designed with a single degree of freedom. This controlled environment allows for precise monitoring and adjustments, ensuring that the exoskeleton’s performance is optimized prior to real-world application.

The control system of this prototype incorporates an EPOS control card [59], a Maxon motor [60,61,62], and a PLC with CANopen communication [63], forming a robust and reliable basis for precise motor control. This configuration was chosen based on the proven performance of the motor and its control board in similar applications, including its high torque-to-size ratio, which is crucial for powering the lower limb exoskeleton in a compact and efficient manner. For example, Maxon motors are widely used in rehabilitation exoskeletons and prosthetics [60,61,62], where they provide the necessary torque while maintaining a lightweight design. Likewise, whether in medical technology, industrial automation, aerospace or mobility, the precision of Maxon electric drives [34] demonstrate their ability to provide necessary torque and precision in a compact form factor, essential for handling heavy loads or precise movements in restricted spaces. The compact size and high motor performance ensure that the exoskeleton can assist with natural movements and provide effective rehabilitation without being cumbersome. The PLC, specifically the Schneider TM241CEC24T model, Schneider Electric, Elche, Spain, was selected for its wide use in industrial applications, ease of integration, and programming flexibility, which is vital in a research environment where adaptability to different experimental conditions is required. For example, this model is commonly used in automation systems for manufacturing processes, where its robust performance and flexibility facilitate the management of complex machines and production lines. Furthermore, it is used in smart building systems to control various subsystems such as lighting, HVAC, and security, demonstrating its versatility and reliability in diverse and demanding scenarios. This broad application base underscores its suitability for controlling prototype exoskeletons, where precise and adaptive control is essential for effective testing and development. This PLC is not only capable of controlling the motor torque with high precision but is also an integral part of managing the overall motion dynamics of the prototype. Its ability to control up to 63 devices on your CANopen network in slave mode offers significant scalability, allowing multiple exoskeleton units or other devices to be managed simultaneously by a single PLC. This feature is particularly beneficial for expanding system capabilities in more complex rehabilitation configurations or for integrating additional sensors and actuators into the control loop. In addition, it has five communication ports, which allows it to be connected anywhere via Ethernet, wireless access, and web servers to simplify the integration and maintenance of machines.

The compatibility of the Maxon engine with its EPOS controllers and its proven track record in research carried out at the Miguel Hernández University (UMH) further justified its selection [1,64,65,66,67]. Although other types of motors, such as stepper motors, could potentially be employed due to their simplicity and accuracy at low speeds, the Maxon motor’s superior reliability, smoother operation, and precise control over a wide range of speeds led to it being the preferred choice for this application.The PLC controls the torque of the Maxon motor by sending and receiving data via the CANopen network from the EPOS controller to control the desired position of the motor. The desired position is entered into the PLC software EcoStruxure Machine Expert V1.2, which calculates the torque necessary to take the prototype to the desired position. The torque value is sent to EPOS, which produces the torque calculated by the PLC by its PID block in the motor. In turn, the EPOS sends position data to the PLC continuously so that the PLC can recalculate the torque that must be applied at each instant. In Figure 5, we can see a PID block that has as inputs ACTUAL, SET_POINT, KP, TN, TV, Y_MANUAL, Y_OFFSET, Y_MIN, Y_MAX, MANUAL, and RESET. The ACTUAL input receives the current position of the prototype sent by the EPOS card; the SET_POINT input receives the desired angular value of the movement; the KP, TN, and TV inputs receive the configuration of the PID controller; and the Y_MIN and Y_MAX inputs receive the minimum and maximum torque values from which the PID controller acts on the prototype. If you want to perform manual control, you can use the Y_MANUAL input to define the controller output and actuate the MANUAL input. This was not an option that we wanted as we were interested in automatic control of the prototype. As for the outputs, the block provides the Y output, which receives the calculated torque value that will be sent to the EPOS, and as information outputs, it has LIMITS_ACTIVE in the case of acting on a limit value, whether higher or lower, and OVERFLOW.

Initial testing involved using a PID controller to manage the prototype’s pendulum-like motion, with parameter adjustments being performed through trial and error to ensure smooth and controlled movements. The results, as illustrated in Figure 6, demonstrated variability in performance across different angles for two different PID controller configurations. Specifically, the graphs in Figure 6 show how the PID controller’s response varied with respect to the setpoint (orange) and the actual position (blue), highlighting fluctuations in control accuracy and stability at different angular positions. It is also clear that the same PID controller configuration does not maintain the same performance for different angular positions. Another important point that was analyzed is the fact that the start of the movement produced by the PID controller may be a little abrupt to be carried out on a person undergoing rehabilitation.

To try to improve this performance, a fuzzy controller was integrated into the system. This integration required the development of fuzzy logic blocks in accordance with the IEC 61131-7 standard, which provides a structured approach for implementing fuzzy control in industrial automation systems. The design process involved the creation of several main function blocks to handle various aspects of fuzzy logic processing, including the implementation of membership functions [68].

The first function is the fuzzification block This block has as inputs the “ERROR” signal and nine additional inputs (X1 to X9) along with an adjustment parameter (AP) for trapezoidal functions. These inputs correspond to the values that are used to create the membership functions, triangular and trapezoidal functions. The membership functions were created in Function Block with ST (Structure Text) language, where they receive the input values from X1 to X9 and calculate the waveform, triangular or trapezoidal, according to the chosen input values. We chose to work with trapezoidal functions at the ends and triangular functions for the remaining functions. After selecting the input values, the fuzzyfication block “calls” the membership functions to calculate their value in relation to the ERROR input. Membership functions play a crucial role in Fuzzy logic by defining how input values are mapped into degrees of membership within various fuzzy sets. In this case, the functions calculate the output value which can be one of the nine fuzzy sets: “Negative_Large_Large”, “Negative_Large”, “Negative_Mediun”, “Negative_Small”, “Zero”, “Positive_Small”, “Positive_Medium”, Positive_Large”, and “Positive_Large_Large”. Each fuzzy set corresponds to a range of error values, with the membership function quantifying the degree to which the error value belongs to each set.

The outputs of the fuzzyfication block, from MD1 to MD9, correspond to the control actions determined by the fuzzy inference process. MD1 corresponds to the “Negative_Large_Large” condition, MD2 to the “Negative_Large” condition, and so on, up to MD9 corresponding to the “Positive_Large_Large” condition. Additionally, the function block provided outputs for “Error” and “Error_MSG” signals. The “Error” output is INT data type and indicates the error code, while the “Error_MSG” is STRING data type and describes the error. Its use is optional. All other entries and outputs of the fuzzyfication block are REAL data type.

The outputs from MD1 to MD9 of the fuzzyfication block are the inputs used by the function blocks of the fuzzy controller rules. The rules created were for errors “Negative_Large_Large” to VNGG response, “Negative_Large” to VNG response, “Negative_Medium” to VNM response, “Negative_Small” to VNP response, “Zero” to VNULO response, “Positive_Small” to VPP response, “Positive_Medium” to VPM response, “Positive_Large” to VPG response, and “Positive_Large_Large” to VPGG response.

The use of membership functions enabled the fuzzy controller to interpret the inputs in a nuanced manner, allowing for more adaptive and precise control compared with traditional PID controllers. By mapping input values to fuzzy sets and using inference rules, the controller could handle the system’s variability and uncertainties more effectively, leading to improved performance and reliability of the prototype.

The rule responses are part of the inputs of the defuzzyfication block. The defuzzyfication block has two sets of inputs. The first set of entries contains the already mentioned responses to the rules that are in the entries named MDX1 to MDX9, with VNGG in the MDX1 entry, VNG in the MDX2 entry, and so on up to VPGG in the MDX9 entry. The other set of inputs corresponds to inputs X1 to X9 of the block. These inputs are associated with the inputs “Negative_Torque_Large_Large” in X1, “Negative_Torque_Large” in X2, “Negative_Torque_Medium” in X3, “Positive_Torque_Large” in X8, and “Positive_Torque_Large_Large” in X9. The values of the inputs from X1 to X9 are the torque values chosen by the designer. The association of these entries is performed by relating X1 with MDX1, X2 with MDX2, and so on up to relating X9 with MDX9. For the output function, the center of gravity method for simple and discrete elements (CoGS) was used (2). The output of the block is the torque value to be sent to the EPOS. All data in this block, inputs and outputs, are REAL type. Figure 7 shows the fuzzyfication block, the membership functions, and the defuzzyfication block created.

To facilitate the management of the control system, an HMI (human–machine interface) was developed where all the data necessary for the tests can be entered. This HMI was created in the same PLC programming software, Machine Expert V1.2. In this HMI, the system is initialized, and the PID controller parameters are defined, which are the KP, TN, and TV values as well as the maximum and minimum values that the controller can act on. The input parameters of the fuzzyfication block are also inserted, parameters from X1 to X9, as well as the parameters of the defuzzyfication block “Negative_Torque_Large_Large” up to “Positive_Torque_Large_Large”. We also insert the number of desired repetitions, the desired setpoint, the weighting between the controllers, and some parameters that can be monitored in real time, such as movement time, applied torque, current torque, current position, PID and FUZZY weighting values, and torques weighted between PID and FUZZY, as shown Figure 8.

In Figure 9, the control loop is shown; with the HMI that sends and receives information to the PLC, in turn, the PLC that sends the desired torque value to the EPOS, which transmits the torque to the motor that returns the position to the EPOS. In turn, the EPOS returns the position information received from the motor to the PLC. The motor moves the prototype to reach the desired position.

### 3.3. Controller Performance and Optimization

Initial results with the PID controller revealed a controller that was able to stabilize the prototype but had sudden movements when changing trajectory.

The results with the fuzzy controller showed some challenges, including delayed stabilization and inconsistent performance. Despite this, it had a better performance at the beginning of the movement than the PID controller, smoother and more continuous. This led to the exploration of combining PID and fuzzy controllers to leverage their respective strengths. The first attempt at combining these controllers involved simple weighting ratios, as illustrated in Figure 10 (left side). The idea of weighting is to change controllers as we approach the setpoint, starting with a weight of 100% of the fuzzy controller, due to its initial movement being less aggressive than the initial movement of the PID controller, and progressively changing the weight of the fuzzy controller by the PID controller weight until the action is 100% of the PID controller, which proved to be better in the final stabilization of the movement. This approach showed improvements in achieving smoother and more stable movements by utilizing fuzzy control for initiation and PID control for stabilization.

Further optimization involved adjusting the weighting curve to a more complex function (Figure 10, right side), which was implemented through a custom block in Structured Text (ST) programming (Figure 11).

The inputs of the weighting block are “INPUT_ERROR”, “DIVISOR_POND”, “SETPOINT”, “PERCENTUAL”, “COEFFICIENT_FUZZY”, “COEFFICIENT_SUM”, and “ERROR”. The outputs are “PONDERACAO_FUZZY” and “PONDERACAO_PID”. The input “ENTRADA_ERRO” receives the positive value of the error. The “DIVISOR_POND” input receives the positive value of the difference between the setpoint and the current position at the moment the chosen percentage value is reached, while the “SETPOINT” input receives the positive value of the difference between the setpoint and the current position at the moment the movement start instruction is sent. The “Percentage” entry receives the value of the weighting that you want to work on. The inputs “COEFFICIENT_FUZZY” and “COEFFICIENT_SUM” receive the values of the coefficient from the weighting equation. The entry “ERROR” is the actual value of the error. When it receives the input values, the block will perform the weighting calculations, and the corresponding weighting value of the fuzzy controller will be at the “PONDERACAO_FUZZY” output, while the corresponding weighting value of the PID controller will be at the “PONDERACAO_PID” output. As the control is now performed with the action of the two controllers together, the torque value sent by the PLC to the EPOS controller will be the sum of the torque weights of the two controllers, fuzzy and PID, starting with 100% of the torque of the fuzzy controller, which over time will decrease its percentage of action while increasing the percentage of action of the PID controller to the point where the PID controller will have control and stabilize the system.

Another important implementation was to create a ramp for the fuzzy controller’s response. This is because, although the fuzzy controller demonstrated that its initial movement was smoother than that of the PID controller, as this is a project that aims to assist rehabilitation patients, we would have to ensure that the response followed a desired path. Figure 12 shows the ramp block created. The inputs are “torque_defuzz”, which is the output of the defuzzyifier block. The ASCEND and DESCEND inputs, which are the values that we will choose as progressive and regressive for the Fuzzy controller and the TIMEBASE input, will regulate how often we will allow the output to be increased or decreased with the ASCEND and DESCEND values. The block also has a RESET input. As an output, we have “Fuzzy_ramp”, which will be our value used by the controller to ensure a smoother movement.

The initial tests were carried out with the weighting curve without percentage variation, which means that the weighting occurred in 100% of the movement. The next idea was to work with weighting after certain percentages of the movement had already been completed. After this percentage is completed, weighting begins and continues until the end of the movement. The beginning of the movement is always started with the action of the fuzzy controller.

Testing various weighting percentages has provided insights into how these settings impact control performance based on different angles of movement within the system. Each test was designed to assess the effectiveness and robustness of the control system for a range of angles, revealing how different weightings influence performance under various conditions.

These findings indicate that adjusting the weighting of PID and fuzzy controllers according to the specific movement angle can optimize overall system performance. By fine-tuning the balance between PID and fuzzy control based on the movement requirements, the system can achieve more precise and stable control. Here is a more detailed breakdown of the performance for each weighting configuration:

**A 70% weighted controller:** This configuration proved to be highly effective for movements of 30 degrees, as depicted in Figure 13a. It demonstrated consistent performance and robustness within the range of 25 (Figure 13b) to 35 degrees (Figure 13c). This implies that the 70% weighting strikes a good balance for moderate movements, making it suitable for applications where stability and smooth control are critical within this angle range.

**A 50% weighted controller:** The 50% weighting controller was found to be adequate for movements of 20 degrees (Figure 13d). It displayed effective robustness in the angle range of 17 (Figure 13e) to 24 degrees (Figure 13f). This suggests that this weighting is better suited for smaller amplitude movements, where precise control is needed in a narrower angle range.

**A 30% weighted controller:** This controller configuration was effective for movements of 40 degrees (Figure 13g). It showed robustness across a broader angle range, specifically between 35 and 44 degrees, illustrated in Figure 13h,i, respectively. This indicates that the 30% weighting is more effective for larger amplitude movements, offering better control and stability for wider movement ranges compared with the higher-percentage configurations.

**A 100% weighted controller:** The 100% weighting configuration was found to be the most suitable for movements in the range of 45 to 50 degrees, as shown in Figure 13j,13k, respectively. Its performance in this range indicates that it excels in handling extensive movements, providing optimal control for larger angles.

In summary, the results suggest that a tailored combination of PID and fuzzy controllers, adjusted based on specific movement angles, can significantly enhance system performance. Controllers with different weighting percentages are better suited to different movement ranges, with the 70% weighting being ideal for moderate movements and the 100% weighting being optimal for extensive movements. To further improve control in exoskeleton systems, ongoing research and testing are necessary to refine these combinations and enhance their adaptability to various conditions and operational requirements.

The results obtained through Table 1 were checked. In Table 1, each graph shows the desired value, the value at which stabilization occurred, the error in degrees, the percentage error, and the time that the system took since the initial request for movement until the moment the system stabilizes. It can be seen that all results are satisfactory, with all errors being below 5% and almost all of them being below 3%.

The data presented in Table 1 further reveal a high degree of accuracy and responsiveness in the system. For instance, in graph (a), the desired value of 30° was achieved with a stabilization value of 29.590°, resulting in an angular error of just 0.41° and a percentage error of 1.39%, with a stabilization time of 4.57 s. Similarly, graph (b) shows that a desired angle of 25° was stabilized at 24.37°, with an angular error of 0.63° and a percentage error of 2.59%, taking 4.3 s to stabilize.

Other notable results include graph (e), where a target angle of 17° was reached with a very minimal error of 0.13°, translating to a percentage error of only 0.76%, and where it achieved stabilization in just 1.64 s. In contrast, graph (i) demonstrates a larger error for a 44° target, where the system stabilized at 45.67°, leading to an angular error of 1.67° and a percentage error of 3.66% with a stabilization time of 4.9 s.

Table 2 shows the parameters used to adjust the controllers. When the operator selects the movement and percentage, the parameters are automatically transferred to the PID and FLC controllers.

### 3.4. Quantum Computing Analysis

The complex application of the aquatic exoskeleton conceives problems with the tuning problem of PID and fuzzy controllers. For this reason, proper tuning of these controllers determines the system’s accuracy and efficiency. In PID controllers, the proportional, integral, and derivative gains (Kp, Ki, Kd) provide adjustment of the system’s response. An incorrect Kp value can lead to oscillations or slow responses, while Ki provides the elimination of the steady-state error and Kd helps to dampen the response. However, manually tuning these parameters is complicated, especially in nonlinear systems or systems with changing dynamics, as the previous results have shown.

On the other hand, tuning FLCs involve adjusting the membership functions and inference rules that define how the system responds to different conditions. Membership functions allow for modeling uncertainty and vagueness in input signals, which is crucial in variable environments. However, adjusting these functions and rules to achieve optimal response can be a complex process, requiring deep knowledge of the system and often needing advanced optimization tools.

Moreover, the need to optimize the weighting between PID and fuzzy controllers adds another layer of complexity. Combining both approaches can leverage the precision of PID for stabilization and the adaptability of fuzzy to handle nonlinearities and variations.

Quantum computing offers significant potential for optimizing PID and fuzzy controllers in real time. By leveraging quantum algorithms like the quantum approximate optimization algorithm (QAOA) and quantum Fourier transform (QFT), system parameters can be dynamically fine-tuned. These algorithms enable the exploration of a wide range of configurations, ensuring optimal controller gains while maintaining system stability and minimizing errors. QAOA iteratively adjusts parameters, while QFT analyzes the results, providing deeper insights into configurations that reduce control errors, making them ideal for complex optimization tasks in dynamic systems like exoskeleton control. However, due to current limitations in quantum computing, such as qubit coherence, error rates, and the complexity of quantum hardware, this proposal remains theoretical at this stage.

In this proposal, the optimization of this weighting according to movement angles could facilitate balance between smoothness and stability in exoskeleton control. This challenge justifies the need for advanced tools, such as QC, which can perform these optimizations more efficiently and in real time.

Therefore, improving the tuning of PID and fuzzy controllers involves optimizing parameters such as the proportional, integral, and derivative gains (Kp, Ki, Kd) of the PID controller, as well as the membership functions and inference rules of the fuzzy controller. Additionally, the weighting between the two controllers is optimized according to the movement angles. To achieve this, a theoretical model with quantum logic gates is used, where the controller parameters (Kp, Ki, Kd, membership functions) are represented as quantum states in qubits. Each qubit can be in a superposition of states representing different possible values of the parameters.

For this proposal, Kp, Ki, and Kd were obtained for the previous results of controller performance, and optimization regarding the values for the PID parameters Kp (proportional gain) is indicated in Table 2, with specific values for different weighting configurations (100% weighting: Kp = 0.015; 70% weighting: Kp = 0.015; 50% weighting: Kp = 0.016 and; 30% weighting: Kp = 0.015).

Moreover, optimal Ki and Kd gains need to be adjusted and calculated during the quantum modeling phase in order to optimize the PID controller performance in a reiterative way. By leveraging the quantum computing framework, these gains can be fine-tuned using quantum algorithms such as the quantum approximate optimization algorithm (QAOA). This would allow for dynamic optimization in real time, improving system stability and reducing control errors. Through techniques like QAOA and quantum Fourier transform (QFT), the system could explore a wide range of configurations, ensuring that the optimal set of gains is consistently applied for varying operational conditions.

The proposed quantum circuit will include quantum gates to manipulate these states and find the optimal combination, specifically:▪**Hadamard gates (H)**: to create a superposition of all possible states.▪● and ⊕ represent the **CNOT gates** between adjacent qubits: to entangle qubits and account for interactions between different parameters.▪**Rotational gates (RZ and RY)**: to adjust the probability of each state, optimizing the membership functions and PID gains.▪**M:** the measurement of each qubit.

Finally, the quantum approximate optimization algorithm (QAOA) will be applied to iteratively adjust the controller parameters in search of the optimal configuration, and the quantum Fourier transform (QFT) will be used to identify the parameter combinations that minimize the control error as the following, where:▪**H**: applies Hadamard gates to the qubits in the QFT.▪**●** and ⊕: represents controlled gates applied in the QFT to achieve entanglement and phase adjustments.▪**RZ(αi)**: represents rotation gates used in the QFT to perform phase shifts.

This block represents a QFT that can be used to process the information obtained from the QAOA. Although the QFT does not directly adjust the QAOA parameters, it provides a means to analyze the results of the QAOA and can help identify the optimal configurations that minimize control error. Thus, the combination of QAOA and QFT methods can be effective for solving complex combinatorial optimization problems and analyzing the obtained results.

According to these parameters, the following six blocks are included:(1)Initial state preparation block:q0: —|0>—┤q1: —|0>—┤q2: —|0>—┤q3: —|0>—┤(2)Hadamard gate application block:q0: —H—┤q1: —H—┤q2: —H—┤q3: —H—┤(3)Entanglement block (CNOT gates):q0: — ● —————┤     │q1: —⊕— ● ———┤       │q2: ———⊕— ● —┤         │q3: —————⊕ —┤(4)Rotational gate application block:q0: —RZ(θ1)—RX(θ2)—┤q1: —RZ(θ3)—RY(θ4)—┤q2: —RZ(θ5)—RY(θ6)—┤q3: —RZ(θ7)—RY(θ8)—┤(5)Quantum Fourier Transform (QFT) block:q0: —H —●—RZ(α1)—┤       │q1: —H—⊕—RZ(α2)—┤       │q2: —H—⊕—RZ(α3)—┤       │q3: —H—⊕—RZ(α4)—┤(6)Measurement block:q0: —————Mq1: —————Mq2: —————Mq3: —————M

## 4. Discussion

This work proposes the development of the exoskeleton prototype focused on aquatic rehabilitation, specifically targeting lower limbs by leveraging water properties such as buoyancy and resistance. A Maxon motor, together with an EPOS control card and a Schneider TM241CEC24T PLC, Schneider Electric, Elche, Spain., was selected for its reliability and precision in movement control. Initial tests using a PID controller showed variability in control accuracy depending on the angle of movement. Subsequently, a fuzzy controller was integrated to improve performance, adjusting membership functions and inference rules.

The initial results of the PID controller were able to stabilize the prototype but with abrupt movements. By combining PID with fuzzy logic, the aim was to smooth the initial movement and stabilize it at the end. Tests with different weightings between the controllers showed that specific adjustments based on the movement angle significantly improved performance. Weightings of 70% and 100% were optimal for moderate to wide movement amplitudes, respectively. This suggests that a tailored combination of both controllers, adjusted according to the range of movement, can optimize the performance of the rehabilitation system.

Traditional methods like PID controllers remain relevant, especially in single-input single-output (SISO) systems. However, their effectiveness decreases in highly nonlinear systems, requiring frequent adjustments. On the other hand, fuzzy logic controllers are more effective at managing multiple input variables and nonlinear systems due to their ability to process inputs in a way that mimics human reasoning. The integration of neural networks with fuzzy logic has shown to improve adaptive control, although PID controllers are still used in combination with modern techniques to manage uncertainties and unmodeled dynamics.

The results of the study on the control of the exoskeleton controlled by a PLC using PID controllers, fuzzy controllers, and a combination of their techniques provide valuable insights into performance optimization. PID controllers proved effective for movements under stable conditions, but their performance decreased in the presence of high nonlinearity. On the other hand, fuzzy controllers demonstrated greater capacity to handle nonlinear systems and adapt to variations, although they faced challenges with stabilization and response time. Overall, the results indicate that the system performs efficiently across various scenarios, maintaining errors within acceptable limits and demonstrating reliable performance with reasonable stabilization times. This suggests that the control algorithms are well tuned to manage the system’s response to different movement requests effectively.

However, the combination of PID and fuzzy controllers turned out to be a promising solution, leveraging the PID controller’s precision for stabilization and the fuzzy controller’s adaptability for initiating movements. Adjusting the weighting between the two methods optimized performance according to the movement angle, enhancing both smoothness and stability. The tests showed that this combination can offer more precise and adaptable control for rehabilitation applications.

Robustness tests indicated that the combined system is effective within ranges close to those studied, but there is still room for improvement in managing noise and disturbances. The next phase of research should focus on refining the algorithms and exploring new strategies to optimize control under more extreme conditions, thus continuing the advancement in exoskeleton technology.

This integrated method, which combines PID and fuzzy control with sophisticated weighting strategies, not only improves the fluidity and stability of movements but also guarantees consistent performance of the exoskeleton across various conditions. The outcome is a highly advanced and robust control system that represents a major advancement in exoskeleton technology, especially for applications demanding both accuracy and adaptability. Moreover, future applications suggest other possible combinations such as remotely operated vehicles (ROVs) for complex underwater systems [69] or model predictive controls (MPCs) based on active disturbance rejection control (ADRC) [70], which have been demonstrated to be efficient for underwater complex conditions, potentially leading to even more sophisticated and adaptable control solutions in challenging environments where aquatic rehabilitation exoskeletons could be utilized.

### 4.1. Challenges and Advances in Exoskeleton Control and Customization with Quantum Computing

Traditional control systems such as PID and fuzzy often struggle to handle the complexity and nonlinearity in real time, especially in dynamic systems like exoskeletons. Additionally, classical control models may be constrained by computational capacity when processing and analyzing large amounts of sensor data in real time. Lastly, exoskeletons face challenges in managing multiple variables and constraints simultaneously, which can lead to suboptimal decisions or frequent manual adjustments.

The complex application of the aquatic exoskeleton faces challenges with tuning PIDs and FLCs [57]. Proper tuning of these controllers is essential for the system’s accuracy and efficiency. In PID controllers, adjustments to proportional (Kp), integral (Ki), and derivative (Kd) gains determine the system’s response. An incorrect Kp value can lead to oscillations or slow responses, while Ki helps eliminate steady-state error and Kd helps to dampen the response. However, manually tuning these parameters is complicated, especially in nonlinear systems or in systems with changing dynamics.

On the other hand, tuning FLC involves adjusting membership functions and inference rules that define how the system responds to different conditions [71]. Membership functions model uncertainty and vagueness in input signals, which could be implemented in variable environments. Adjusting these functions and rules to achieve an optimal response can be a complex process, requiring deep knowledge of the system and often advanced optimization tools.

The need to optimize the weighting between PID and fuzzy controllers adds another layer of complexity. Combining both approaches can leverage the precision of PID for stabilization and the adaptability of fuzzy to handle nonlinearities and variations. Optimizing this weighting according to movement angles is crucial for balancing smoothness and stability in exoskeleton control. This challenge justifies the use of advanced tools, such as quantum computing, which can perform these optimizations more efficiently and in real time.

Improving the tuning of PID and fuzzy controllers involves optimizing parameters such as the proportional, integral, and derivative gains (Kp, Ki, Kd) of the PID controller, as well as the membership functions and inference rules of the Fuzzy controller. Additionally, the weighting between the two controllers is optimized according to the movement angles. A theoretical model using quantum logic gates is employed, where the controller parameters (Kp, Ki, Kd, membership functions) are represented as quantum states in qubits. Each qubit can be in a superposition of states representing different possible values of the parameters.

The proposed quantum circuit will include quantum gates to manipulate these states and find the optimal combination, specifically: (1) Hadamard gates will create a superposition of all possible states; (2) CNOT gates will entangle qubits and account for interactions between different parameter; (3) rotational gates will adjust the probability of each state, optimizing the membership functions and PID gains; (4) there will be measurement of each qubit; and (5) the QAOA will be applied to iteratively adjust the controller parameters in search of the optimal configuration, and the QFT will be used to identify the parameter combinations that minimize the control error. The combination of QAOA and QFT methods can be effective for solving complex combinatorial optimization problems and analyzing the obtained results.

Quantum computing can transform modeling and prediction in lower limb exoskeleton systems by enabling the processing of large data volumes with unprecedented efficiency [72]. Quantum algorithms for machine learning can analyze complex patterns and user behaviors with much greater accuracy than classical models. This advanced analytical capability facilitates the creation of more precise predictive models, allowing for finer adaptation to specific user conditions and requirements. As a result, the exoskeleton can offer a more effective personalized experience, adjusting to different tasks and environments more efficiently.

Regarding real-time optimization problem management, quantum computing provides a significant advantage by solving complex issues with multiple variables and constraints simultaneously. Quantum algorithms can handle these problems more effectively than traditional methods, allowing for instant adaptation to changes in the environment or user state. This quick and precise response capability reduces the need for human intervention and improves the overall efficiency of the exoskeleton, making the device more reliable and versatile in dynamic situations. Finally, quantum computing can accelerate and enhance the exoskeleton customization process. The ability to process large data volumes more rapidly enables more efficient training of AI models, facilitating rapid adaptation to each user’s specific needs. Quantum learning algorithms can fine-tune the exoskeleton’s behavior with greater precision, offering a more customized and effective solution. This approach not only improves the exoskeleton’s functionality but also provides a more effective solution for rehabilitation and assistive applications, making the devices more accessible and useful for a wide range of users.

On the other hand, it is important to note that, as a prototype in the initial testing phase, user trials have not yet been conducted. However, the control system is being designed with mechanisms that will allow for adjustments to support and movement speed based on the user’s strength and endurance in future development phases.

The system design integrates algorithms that enable dynamic adjustment of motor torque in response to the resistance demands specified by the user. These adjustments are based on sensor readings that continuously monitor the user’s physical state. In future user trials, these data will be used to adjust resistance and support in real time. For example, the system will be able to automatically reduce resistance based on detected fatigue or increase support to compensate for a decrease in the user’s strength.

Additionally, during future development, fine-tuning of the control algorithms is planned to optimize system personalization according to each user’s individual characteristics. This will allow for a more tailored experience to the needs of each individual and improve the system’s effectiveness in real-world scenarios. Validation with real users will be a crucial step in refining these mechanisms and ensuring that the system can effectively adjust support and movement speed based on the user’s strength and endurance.

As technology continues to advance, it is likely that we will see even broader adoption of these devices across various aspects of daily life, transforming the way we approach mobility and rehabilitation. The long-term impact of rehabilitation technologies, such as wearable robotics, which also includes aquatic therapy by exoskeletons, can be substantial for potential clinical outcomes patient rehabilitation [73], and it could impact therapists’ perspectives [74]. In terms of rehabilitation, the prolonged use of exoskeletons can lead to lasting improvements in muscle strength, mobility, and autonomy. For example, regular use of exoskeletons, such as ReWalk [11] and EksoGT [9], has been demonstrated to enable patients to perform repetitive movements and monitor their own progress, leading to continuous improvements in motor function and physical endurance.

Aquatic therapy, on the other hand, provides long-term benefits by alleviating chronic pain, improving flexibility, and reducing joint impact. The buoyancy of water decreases the load on the body, allowing patients to perform movements that would be difficult or impossible on land. This is particularly beneficial for individuals with arthritis, chronic back pain, or those recovering from surgery. The pain reduction and mobility improvements experienced during aquatic therapy can translate into greater participation in daily activities and an overall enhancement in quality of life. The therapeutic benefits of water also extend to mental health, providing a relaxing and less stressful environment for rehabilitation [75].

In addition to physical benefits, the long-term improvement in quality of life also includes emotional and psychological aspects [76]. The independence achieved through the use of exoskeletons and aquatic therapy can lead to increased self-confidence and self-esteem. Patients who previously felt limited by their health conditions can now participate in social and recreational activities with greater ease. This not only improves their overall well-being but also positively impacts their mental and emotional health.

By integrating these advanced technologies into therapeutic protocols, exoskeletons are transforming rehabilitation practices, offering new opportunities for recovery and improved functional independence. The future of rehabilitation appears to be increasingly promising, bringing hope and opportunities for better recovery to people worldwide. The integration of these technologies not only enhances the efficiency of treatments but also offers a more comfortable and less painful experience for patients. 

The use of water properties such as buoyancy and resistance in the exoskeleton provides several advantages. Buoyancy reduces the load on the joints, which helps to alleviate discomfort and allows for a broader range of motion during rehabilitation. Water resistance contributes to a more controlled and progressive resistance profile, which can enhance the effectiveness of exercises and improve muscle strength and endurance over time.

Additionally, the system incorporates advanced control mechanisms that adjust to varying conditions and patient needs in real time. This adaptability ensures that the rehabilitation protocols are more tailored to individual requirements, leading to a more personalized and effective rehabilitation experience. The real-time data collection and monitoring allow for continuous adjustments based on feedback from the system, ensuring that the movements are optimized for both comfort and efficacy. By integrating these features, the system aims to offer a more effective and user-friendly rehabilitation experience, improving overall performance and satisfaction in aquatic environments.

Furthermore, the ability to adjust the control parameters dynamically based on the user’s specific needs enhances the system’s responsiveness. This capability allows for more precise control over the movement and resistance, contributing to a more comfortable and effective rehabilitation process. The detailed real-time data analysis also helps in identifying and addressing any issues promptly, which leads to a smoother and more efficient rehabilitation experience. Overall, these improvements collectively enhance both the functional performance of the system and the user’s overall experience during aquatic rehabilitation.

### 4.2. Limitations of Current Exoskeleton and Aquatic Therapy Technologies

The current prototype of our exoskeleton is designed with a single degree of freedom, focusing on the fundamental aspects of motor control and system performance within a controlled environment. This initial design is intended to establish a baseline for evaluating control strategies and optimizing performance before advancing to more complex configurations. The choice of a single degree of freedom allows for detailed analysis and refinement of the control mechanisms, which is essential for ensuring accuracy and reliability. As the development progresses, it is planned to incorporate additional degrees of freedom to simulate more realistic and complex movements. This will enable a thorough investigation into the interactions between multiple joints and links, addressing the increased complexity required for a fully functional exoskeleton. The ongoing research and future iterations will aim to integrate these elements, ensuring that the final design meets the necessary requirements for comprehensive rehabilitation applications.

One significant limitation of the current study is its focus on classic control methods, such as PID and fuzzy controllers, which may not fully address the complexities associated with nonlinearities and uncertainties in the exoskeleton system. These classic methods provide a foundational understanding but are inherently limited in scenarios where nonlinear behavior and system uncertainties play a critical role. As noted in the literature, handling these factors often requires more advanced control strategies [62,63,64]. This initial analysis of design does not yet incorporate adaptive or robust control methods specifically designed to manage such complexities. Future research will address this limitation by integrating and evaluating advanced control techniques, which will involve both simulation and practical experimentation to better handle nonlinearities and uncertainties in real-world applications.

Despite the advancements in exoskeleton and aquatic therapy technologies, several limitations persist. One significant challenge is the complexity of real-time control in dynamic environments. Traditional PID and fuzzy logic controllers often struggle to handle the nonlinearity and multi-variable nature of exoskeleton systems effectively. These controllers may require frequent manual adjustments to accommodate varying conditions, which can reduce their overall efficiency and responsiveness. Additionally, classical models are often constrained by computational limitations, making it difficult to process and analyze large volumes of sensor data in real time. This can hinder the adaptability of the system, especially in unpredictable or rapidly changing environments.

The emerging field of quantum computing offers a potential solution to these limitations by providing enhanced computational capabilities. Quantum computing could revolutionize the control and optimization of exoskeleton systems by processing complex data and performing high-dimensional optimizations more efficiently than classical systems. Quantum algorithms can tackle intricate control problems involving numerous variables and constraints, potentially enabling real-time adjustments and more sophisticated control strategies. However, the integration of quantum computing into practical applications faces its own set of challenges. These include the current limitations in quantum hardware, the need for specialized knowledge to develop and implement quantum algorithms, and the significant investment required for research and development.

Another limitation is the physical and practical constraints of existing exoskeletons and aquatic therapy devices. For exoskeletons, issues such as device weight, user comfort, and battery life can impact their usability and effectiveness in long-term rehabilitation. Similarly, aquatic therapy equipment can face challenges related to space, water quality, and accessibility, which may limit its availability and convenience for some patients. Both technologies also require precise tuning and calibration, which can be complex and time-consuming. Addressing these limitations involves ongoing research and development to improve the integration of advanced control systems and enhance the overall functionality and user experience of these rehabilitation technologies.

Finally, a potential direction for future research would be to develop monitoring and visualization systems to allow for a more intuitive and direct observation of the exoskeleton behavior during its operation. While all control of the system is managed at the motor level in the present work, it would be beneficial to have additional tools represent the full behavior of the exoskeleton, including not only motor dynamics but also the effects on the structure, joint forces, and patient responses in real time. This approach could involve the use of digital twin models or advanced simulation techniques that provide real-time visualization of the exoskeleton’s behavior, facilitating a more comprehensive assessment of the system and enabling more precise adjustments in control algorithms. Furthermore, these visualization and modeling techniques could also leverage advancements in quantum computing, enabling the implementation of quantum-based optimization methods. By employing advanced quantum modeling, complex optimization problems related to control strategies and exoskeleton performance could be tackled more efficiently, potentially leading to breakthroughs in both exoskeleton design and rehabilitation processes. The integration of these technologies could not only enhance the understanding of exoskeleton performance but also optimize the rehabilitation process by better tailoring it to the patient’s needs.

As the development of the proposed system progresses, it is important to detail the specific advances in rehabilitation and how these improvements are tested. The aquatic exoskeleton system is designed to leverage the properties of water, such as buoyancy and resistance, to enhance range of motion, stretching, and precise, repeatable control. These benefits are evaluated through controlled testing environments using metrics such as range of motion achieved, control accuracy, and effectiveness of rehabilitation protocols. To address real-time monitoring, a data collection system has been implemented to continuously track relevant variables during rehabilitation sessions. This system captures data on movement dynamics, forces applied, and patient responses, providing a comprehensive view of the exoskeleton’s performance and allowing for dynamic adjustments in control. Initial tests have shown that the system can adapt effectively to different conditions and meet individual patient needs, supporting the feasibility of the proposed approach and its potential to improve rehabilitation outcomes.

The current state of the project has not yet included human testing. Future studies should focus on initiating tests with a small participant group to validate control, reliability, and safety when used with patients before reaching the clinical testing phase. These initial trials will involve carefully selected volunteers under strict inclusion and exclusion criteria to ensure safety and compliance with ethical standards. The primary objective will be to verify that the PID-FLC joint control system applied to the human body does not cause any harm while ensuring its stability and effectiveness in controlled conditions. If necessary, other control strategies may be used, such as incorporating adaptive control—suitable for nonlinear systems like this one as it adjusts to varying dynamics and disturbances—or intelligent control strategies, such as neural networks or predictive control. These alternatives may address any limitations encountered during testing. Furthermore, the use of quantum computing is considered a potential differentiator for enhancing the system’s performance. Following these preliminary studies, the next phase will expand the prototype’s degrees of freedom, testing whether the current control techniques remain viable for equipment with more complex functionality and eventually progressing to larger-scale clinical trials to assess real-world effectiveness. It is necessary to verify that the PID-FLC joint control system applied to the human body alone cannot cause any type of harm to the patient. If necessary, other control strategies may be used, such as incorporating a strategy such as adaptive control, indicated for nonlinear systems such as this one, which adapts to varying circumstances of behavior in the dynamics of a system and its disturbances, or intelligent control, such as neural networks or predictive control. As for these strategies, we believe that the use of quantum computing can be a differentiator. The next phase of the study is to expand the prototype’s degrees of freedom and test whether the same control techniques are viable for equipment with more than one degree of freedom.

## 5. Conclusions

This research highlights significant progress in the development of an exoskeleton prototype tailored for aquatic rehabilitation, emphasizing the effective integration of proportional–integral–derivative (PID) and fuzzy logic controllers (FLCs) to enhance control precision and system adaptability. By focusing on lower limb rehabilitation and leveraging water properties such as buoyancy and resistance, the project demonstrates how combining PIDs and FLCs can improve stability and performance. The Maxon motor, EPOS control card, and Schneider TM241CEC24T PLC were selected for their reliability and precision, supporting the refined control mechanisms.

The combined use of these controllers has improved the stability and performance of the prototype, especially for lower limb rehabilitation. Despite these advancements, challenges remain in real-time control and device optimization, such as managing nonlinearities and computational constraints.

Initial tests with the PID controller showed variability in control accuracy based on movement angles, leading to the integration of fuzzy logic to address abrupt movements and improve performance. The combination of PID and FLC was tested with different weightings, revealing that a tailored blend—70% PID for moderate movements and 100% FLC for wider amplitudes—significantly optimized performance. This adjustment enhances both smoothness and stability, underscoring the importance of dynamic control strategies in rehabilitation systems. The traditional PID and FLC approaches, though valuable, often struggle with the complexity and variability inherent in dynamic rehabilitation systems. They require frequent manual adjustments to maintain performance, which can limit their effectiveness and responsiveness. Furthermore, classical computational models face constraints in processing large volumes of real-time sensor data, impacting their ability to adapt swiftly to changing conditions.

For these reasons, real-time control and device optimization remain challenging. Traditional PID and FLC systems, while valuable, often struggle with the complexity and variability inherent in dynamic rehabilitation environments. They require frequent manual adjustments to accommodate nonlinearities and computational constraints, which can limit their effectiveness and responsiveness. Classical computational models also face constraints in processing large volumes of real-time sensor data, impacting their ability to adapt swiftly to changing conditions.

Quantum computing offers a promising solution to these challenges by revolutionizing how we handle control system tuning and optimization. Specifically, quantum computing can greatly enhance the tuning of PID and fuzzy logic controllers by addressing their limitations in handling complex, multi-variable problems. By representing controller parameters—such as proportional (Kp), integral (Ki), and derivative (Kd) gains for PID, and membership functions for fuzzy logic—as quantum states in qubits, quantum computing can explore a vast array of possible parameter combinations simultaneously.

The use of quantum gates, such as Hadamard gates to create superpositions of states, CNOT gates to entangle qubits, and rotational gates to adjust state probabilities, enables precise optimization of these parameters. Quantum algorithms like the quantum approximate optimization algorithm (QAOA) can iteratively refine these parameters to find optimal configurations, while quantum Fourier transform (QFT) aids in identifying parameter combinations that minimize control error. This approach can solve complex combinatorial optimization problems more effectively than classical methods, offering enhanced precision and adaptability in real-time control.

Incorporating quantum computing into exoskeleton development could lead to significant improvements in performance by enabling more accurate and efficient tuning of control systems. This would result in more responsive and adaptable rehabilitation devices, providing tailored solutions that better meet individual patient needs. As quantum computing technology evolves, it promises to further advance the capabilities of rehabilitation technologies, paving the way for more effective and personalized treatments and ultimately transforming the future of mobility and rehabilitation. As research progresses, quantum computing has the potential to revolutionize the field, transforming how we approach mobility and rehabilitation for improved patient outcomes, quality of life, and well-being.

## Figures and Tables

**Figure 1 sensors-24-07765-f001:**
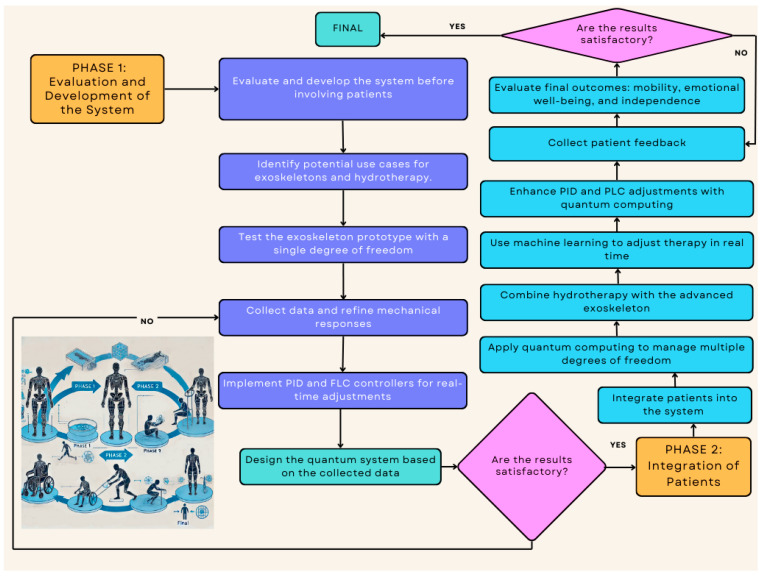
Schematic phases of proposed implementation of aquatic rehabilitation system.

**Figure 2 sensors-24-07765-f002:**
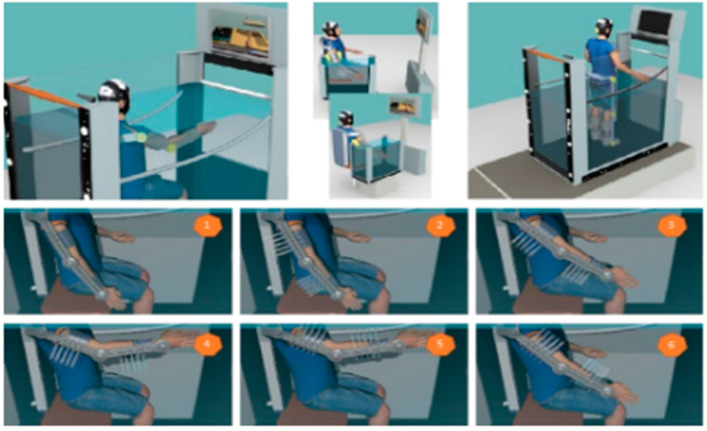
Schematic of the proposed design in the main project NOHA.

**Figure 3 sensors-24-07765-f003:**
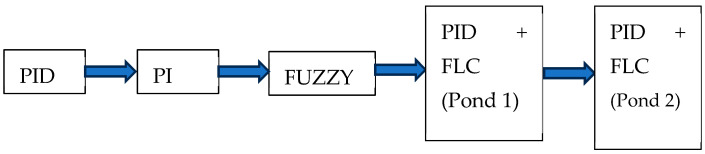
Progression from using the PID method to implementing a hybrid approach that combines PID and fuzzy controllers.

**Figure 4 sensors-24-07765-f004:**
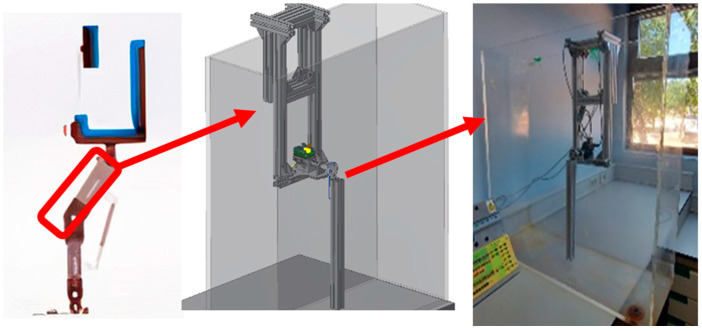
(**Left**) Figure: Lower limb exoskeleton showing the selected part for control. (**Center**) Figure: model created from the selected part of the exoskeleton. (**Right**) Figure: hydrotherapy tank with the prototype created.

**Figure 5 sensors-24-07765-f005:**
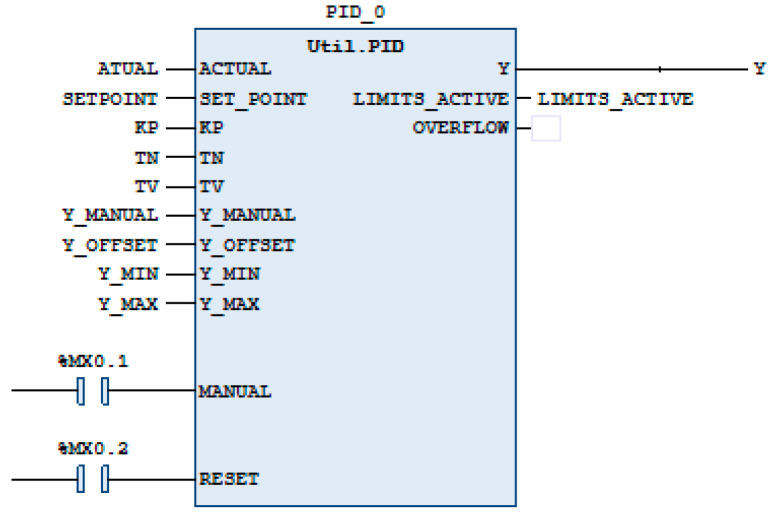
PID block.

**Figure 6 sensors-24-07765-f006:**
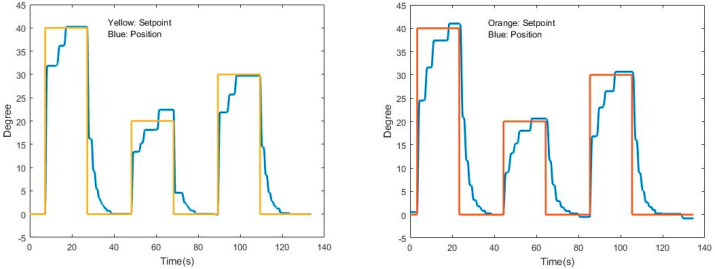
(**Left**): PID graph 1: Degrees x Time (s); yellow is the setpoint and blue is the position. (**Right**): PID graph 2: Degrees x Time (s); orange is the setpoint and blue is the position.

**Figure 7 sensors-24-07765-f007:**
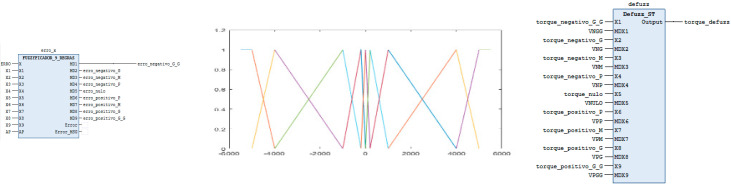
(**Left**): fuzzyfication block. (**Middle**): membership functions. (**Right**): defuzzyfication block.

**Figure 8 sensors-24-07765-f008:**
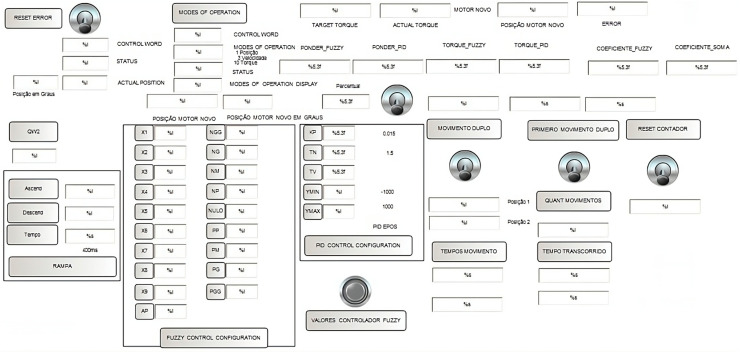
HMI screen created.

**Figure 9 sensors-24-07765-f009:**
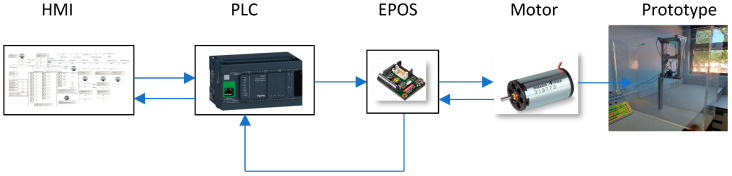
Control loop with the HMI, PLC, EPOS, motor, and prototype.

**Figure 10 sensors-24-07765-f010:**
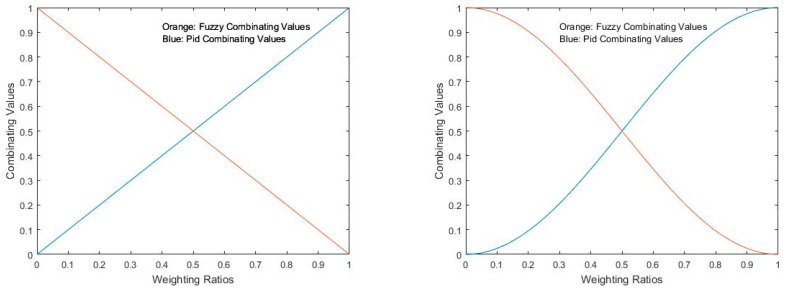
First (**left**) and second (**right**) ponderation graphic.

**Figure 11 sensors-24-07765-f011:**
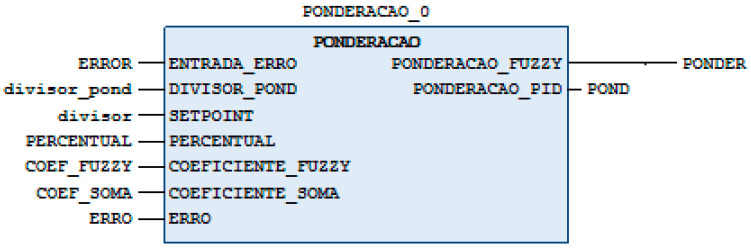
Function block for the ponderation calculations.

**Figure 12 sensors-24-07765-f012:**
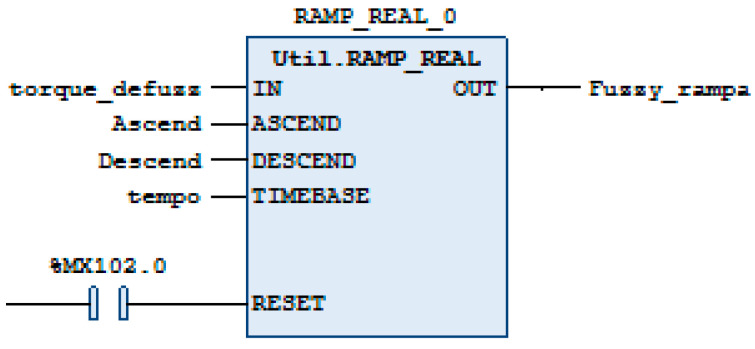
Function block for the ramp block.

**Figure 13 sensors-24-07765-f013:**
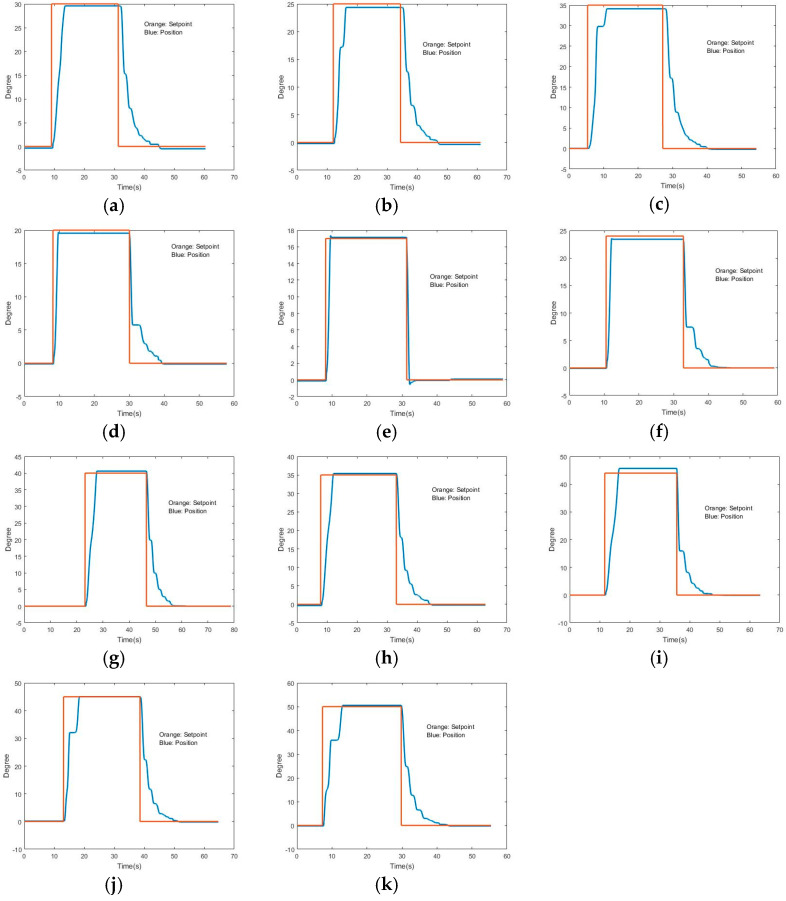
Weighted controller: (**a**). Controller for 70% for 30°. (**b**). Controller for 70% for 25°. (**c**). Controller for 70% for 35°. (**d**). Controller for 50% for 20°. (**e**). Controller for 50% for 17°. (**f**). Controller for 50% for 24°. (**g**). Controller for 30% for 40°. (**h**). Controller for 30% for 35°. (**i**). Controller for 30% for 44°. (**j**). Controller for 100% for 45°. (**k**). Controller for 100% for 50°.

**Table 1 sensors-24-07765-t001:** Numerical results of the graphics.

Graphic	Desired Value (°)	Stabilization Value (°)	Error in Degrees (°)	Percentage Error (%)	Stabilization Time Since Start of Motion Request (s)
a.	30	29.59	0.41	1.39	4.57
b.	25	24.37	0.63	2.59	4.3
c.	35	34.12	0.88	2.58	5.64
d.	20	19.55	0.45	2.30	1.74
e.	17	17.13	0.13	0.76	1.64
f.	24	23.57	0.43	1.82	1.78
g.	40	40.59	0.59	1.45	4.58
h.	35	35.39	0.39	1.10	4.32
i.	44	45.67	1.67	3.66	4.9
j.	45	45.05	0.05	0.11	5.36
k.	50	50.54	0.54	1.07	5.78

**Table 2 sensors-24-07765-t002:** Parameters used for tuning controllers.

ADDRESS	100%	70%	50%	30%
X1	−5000	−5000	−5000	−5000
X2	−4000	−4000	−4000	−4000
X3	−1000	−1000	−1000	−1000
X4	−200	−200	−200	−200
X5	0	0	0	0
X6	200	200	200	200
X7	1000	1000	1000	1000
X8	4000	3000	3000	4000
X9	5000	5000	5000	5000
NEGATIVE_TORQUE_LARGE_LARGE	−50	−150	−50	−100
NEGATIVE_TORQUE_LARGE	−50	0	−50	0
NEGATIVE_TORQUE_MEDIUN	0	0	0	0
NEGATIVE_TORQUE_SMALL	0	0	0	0
ZERO	0	0	0	0
POSITIVE_TORQUE_SMALL	0	0	0	0
POSITIVE_TORQUE_MEDIUN	350	500	800	500
POSITIVE_TORQUE_LARGE	1000	1000	1000	1000
POSITIVE_TORQUE_LARGE_LARGE	1000	600	1000	600
ASCEND	40	30	80	40
DESCEND	40	10	40	40
KP	0.015	0.015	0.016	0.015
TN	1.3	1.3	1.3	1.3
Y_MIN	−1000	−1000	−1000	−1000
Y_MAX	1000	1000	1000	1000
RAMP_TIME	400 ms	400 ms	400 ms	400 ms

## Data Availability

Data are contained within the article.

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
