# Peer review of "Rehabilitation Technologies by Integrating Exoskeletons, Aquatic Therapy, and Quantum Computing for Enhanced Patient Outcomes"

_sensors, 2024, doi:10.3390/s24237765_

Round 1
Reviewer 1 Report (New Reviewer)
Comments and Suggestions for Authors
1. Addressing the Lack of Clinical Testing
The current study does not involve clinical testing on human subjects; instead, it focuses on simulations and controlled laboratory tests. While the paper presents the development of a control strategy for the exoskeleton device and tests a hybrid controller combining PID and FLC, no clinical trials involving human subjects have been conducted in an actual medical environment. To address this limitation, future studies should aim to transition from controlled laboratory settings to clinical trials. A phased approach involving pilot clinical studies with small participant groups can help validate the system’s feasibility and safety, followed by larger-scale clinical trials to assess its efficacy in real-world conditions.
2. Concerns About Verification of Prototype Effectiveness
According to the paper, the current stage focuses on a single-degree-of-freedom control strategy and prototype tested in a laboratory setting, with clinical trials involving human subjects planned for the future. As this is an initial phase of the research, the emphasis is on securing and verifying the stability of the control system and the prototype before conducting actual tests on human subjects. However, to substantiate the prototype's effectiveness, it is crucial to include even a small group of subjects to gather verification data rather than relying solely on development and the authors’ claims. The authors should consider small-scale feasibility tests to collect preliminary data and demonstrate the system’s potential in real clinical scenarios.
3. Limitations of PID and FLC Controllers
The PID controller faces several limitations. It is designed for linear systems, which makes it less effective in handling complex, nonlinear systems like the human body, which involves varying musculoskeletal conditions and temporal changes. PID requires manual tuning of Kp, Ki, and Kd parameters based on the system’s characteristics, and frequent re-tuning may be necessary due to changes in conditions. Furthermore, PID’s reliance on error-based signals can lead to overreactions to noise or abrupt changes, potentially causing instability in controlling human movement.
Fuzzy Logic Controllers (FLC) operate based on rule-based logic but also have drawbacks. Designing FLC requires expert knowledge and experience, making it challenging to systematically establish rules for complex systems like the human body. Fuzzy logic’s dependence on empirical rules and fuzzy sets without a clear mathematical model can lead to unexpected results in complex scenarios. Additionally, real-time computation with numerous input variables can be burdensome, particularly in complex systems like the human body.
When applying PID and FLC to the human body, variations in physiological characteristics and neuromuscular responses among individuals can result in inconsistent control strategies. What works for one individual might not be suitable for another. Moreover, if PID or FLC is not adequately adjusted, excessive force may be applied, leading to injuries. The authors should address these concerns by incorporating adaptive or intelligent control strategies, like model predictive control (MPC) or reinforcement learning, and specify their efforts to mitigate these issues in the discussion section.
4. Addressing the Single-Degree-of-Freedom Limitation
The prototype presented in the study focuses on a single-degree-of-freedom exoskeleton, which limits its application to more complex multi-degree-of-freedom systems. While validating a single-degree-of-freedom system is a meaningful first step, future research must address multi-degree-of-freedom systems. Although the hybrid control strategy of PID and fuzzy control has shown effectiveness within a specific range of angles, its adaptability and capacity to manage complex nonlinear systems remain limited. The authors should discuss plans for future research involving multi-degree-of-freedom systems and enhancing the adaptability of their control strategies in response to changing environmental conditions.
5. Feasibility of Quantum Computing Implementation
While the paper mentions the potential introduction of quantum computing, there is a lack of specific data regarding its practical application benefits. The authors should consider providing empirical evidence or modeling results that demonstrate the feasibility of using quantum computing in this context. Additional research is needed to explore the practical applicability of quantum computing for optimizing control strategies, and this aspect requires more in-depth discussion to bridge the gap between theoretical potential and real-world implementation.
6. Contribution of the Study
This study aims to improve the performance of a robotic exoskeleton system in an aquatic rehabilitation environment through the combination of PID and fuzzy control. Leveraging the physical properties of water to enhance rehabilitation effectiveness and utilizing a hybrid control strategy to improve stability and precision represent a noteworthy technical innovation. The study demonstrates the potential of the proposed approach to maximize rehabilitation benefits through the integration of control theory and aquatic rehabilitation principles.
7. Future Directions
Future studies should focus on extending the system’s applicability to multi-degree-of-freedom exoskeletons and conducting clinical trials involving actual patients to verify the system’s real-world effectiveness. Additionally, the authors should more concretely discuss the feasibility of using quantum computing for control optimization and explore additional strategies to enhance real-time adaptability. Overall, this study makes a significant academic contribution by exploring a novel technological approach to maximizing rehabilitation treatment outcomes. It is anticipated that this foundation will lead to the development of more advanced rehabilitation technologies in the future. Further research is eagerly awaited.
Author Response
- Addressing the Lack of Clinical Testing
The current study does not involve clinical testing on human subjects; instead, it focuses on simulations and controlled laboratory tests. While the paper presents the development of a control strategy for the exoskeleton device and tests a hybrid controller combining PID and FLC, no clinical trials involving human subjects have been conducted in an actual medical environment. To address this limitation, future studies should aim to transition from controlled laboratory settings to clinical trials. A phased approach involving pilot clinical studies with small participant groups can help validate the system’s feasibility and safety, followed by larger-scale clinical trials to assess its efficacy in real-world conditions.
We appreciate very much your review. Thank you very much for your indications. Please, we have added the following paragraph at the end of the discussion chapter:
“The current state of the project has not yet included human testing. Future studies should focus on initiating tests with a small participant group to validate control, reliability, and safety when used with patients before reaching the clinical testing phase. These initial trials will involve carefully selected volunteers under strict inclusion and exclusion criteria to ensure safety and compliance with ethical standards. The primary objective will be to verify that the PID-FLC joint control system applied to the human body does not cause any harm while ensuring its stability and effectiveness in controlled conditions. If necessary, other control strategies may be used, such as incorporating adaptive control—suitable for nonlinear systems like this one as it adjusts to varying dynamics and disturbances—or intelligent control strategies such as neural networks or predictive control. These alternatives may address any limitations encountered during testing. Furthermore, the use of Quantum Computing is considered a potential differentiator for enhancing the system’s performance. Following these preliminary studies, the next phase will expand the prototype's degrees of freedom, testing whether the current control techniques remain viable for equipment with more complex functionality and eventually progressing to larger-scale clinical trials to assess real-world effectiveness.It is necessary to verify that the PID-FLC joint control system applied to the human body alone cannot cause any type of harm to the patient. If necessary, other control strategies may be used, such as incorporating a strategy such as adaptive control, indicated for nonlinear systems such as this one, which adapts to varying circumstances of behavior in the dynamics of a system and its disturbances, or intelligent control, such as neural networks or predictive control. As for these strategies, we believe that the use of Quantum Computing can be a differentiator. The next phase of the study is to expand the prototype's degrees of freedom and test whether the same control techniques are viable for equipment with more than one degree of freedom.”
- Concerns About Verification of Prototype Effectiveness
According to the paper, the current stage focuses on a single-degree-of-freedom control strategy and prototype tested in a laboratory setting, with clinical trials involving human subjects planned for the future. As this is an initial phase of the research, the emphasis is on securing and verifying the stability of the control system and the prototype before conducting actual tests on human subjects. However, to substantiate the prototype's effectiveness, it is crucial to include even a small group of subjects to gather verification data rather than relying solely on development and the authors’ claims. The authors should consider small-scale feasibility tests to collect preliminary data and demonstrate the system’s potential in real clinical scenarios.
Thank you very much for your feedback. We have added the next steps needed for the collection of primary data in humans, which requires the approval of previous functionalities in certified laboratories. As stated in the updated section of the discussion, future studies will focus on initiating tests with a small participant group to validate control, reliability, and safety when used with patients before reaching the clinical testing phase. These initial trials will be conducted under controlled conditions with strict inclusion and exclusion criteria to ensure participant safety. The primary objective will be to verify that the PID-FLC joint control system applied to the human body does not cause harm and is stable in practical use. Following these preliminary studies, larger-scale clinical trials will be planned to evaluate the system's effectiveness in real clinical scenarios. This approach will provide the necessary verification data and substantiate the system's potential for real-world application.
- Limitations of PID and FLC Controllers
The PID controller faces several limitations. It is designed for linear systems, which makes it less effective in handling complex, nonlinear systems like the human body, which involves varying musculoskeletal conditions and temporal changes. PID requires manual tuning of Kp, Ki, and Kd parameters based on the system’s characteristics, and frequent re-tuning may be necessary due to changes in conditions. Furthermore, PID’s reliance on error-based signals can lead to overreactions to noise or abrupt changes, potentially causing instability in controlling human movement.
Fuzzy Logic Controllers (FLC) operate based on rule-based logic but also have drawbacks. Designing FLC requires expert knowledge and experience, making it challenging to systematically establish rules for complex systems like the human body. Fuzzy logic’s dependence on empirical rules and fuzzy sets without a clear mathematical model can lead to unexpected results in complex scenarios. Additionally, real-time computation with numerous input variables can be burdensome, particularly in complex systems like the human body.
When applying PID and FLC to the human body, variations in physiological characteristics and neuromuscular responses among individuals can result in inconsistent control strategies. What works for one individual might not be suitable for another. Moreover, if PID or FLC is not adequately adjusted, excessive force may be applied, leading to injuries. The authors should address these concerns by incorporating adaptive or intelligent control strategies, like model predictive control (MPC) or reinforcement learning, and specify their efforts to mitigate these issues in the discussion section.
Thank you very much for your comments and valuable feedback. We would like to clarify that qualities such as adaptive or intelligent control can be addressed progressively after the initial prototype phase at TRL 3. In this early stage, the primary goal is to validate the technical feasibility of the system and ensure that the basic functionalities of the device, such as control stability and user safety, are properly verified in a controlled environment. Only once these foundations are solidified can we proceed to implement more complex control strategies.
One of the common challenges in the development of exoskeletons is precisely attempting to apply complex adjustments in the early stages and conducting tests in complex environments without first validating the basic functionalities of the system. This can lead to failures, as the device may not adapt appropriately to the complexities of the human musculoskeletal system. The human musculoskeletal system is highly intricate, with significant variations between individuals, and involves dynamic, nonlinear interactions that are not easily modeled or predicted. The complexity lies in how muscles, tendons, bones, and joints work together, responding to both voluntary and involuntary movements, as well as varying forces and external disturbances. This makes it essential to fully understand the system's basic behavior and ensure that it functions properly before testing more advanced adjustments. Without first verifying these basic functionalities, the risks increase, and the device may not perform as expected in real-world scenarios, potentially causing discomfort.
Therefore, these advanced adjustments will be implemented in later stages, once the basic functionalities have been validated. Instead, the integration of advanced technologies will be part of the device’s evolution in later phases, once the prototype has demonstrated stability under controlled conditions and has passed more rigorous testing in complex environments, such as those involving interaction with the human body. We once again appreciate your observations and understanding, and we remain committed to advancing the system’s development in a responsible and progressive manner.
- Addressing the Single-Degree-of-Freedom Limitation
The prototype presented in the study focuses on a single-degree-of-freedom exoskeleton, which limits its application to more complex multi-degree-of-freedom systems. While validating a single-degree-of-freedom system is a meaningful first step, future research must address multi-degree-of-freedom systems. Although the hybrid control strategy of PID and fuzzy control has shown effectiveness within a specific range of angles, its adaptability and capacity to manage complex nonlinear systems remain limited. The authors should discuss plans for future research involving multi-degree-of-freedom systems and enhancing the adaptability of their control strategies in response to changing environmental conditions.
Thank you for your insightful feedback and for emphasizing the importance of addressing multi-degree-of-freedom systems and control adaptability in future research. We would like to point out that this consideration was already incorporated in the previous version of the manuscript, specifically at the beginning of Section 4.2 on limitations. In that section, we highlighted that the current single-degree-of-freedom prototype serves as a foundational step toward the development of more complex systems. We also outlined our plans to incorporate additional degrees of freedom to simulate realistic and complex movements, investigate interactions between multiple joints, and adapt control strategies for nonlinear and dynamic conditions.
We appreciate your suggestion, as it reinforces the importance of these aspects, and we remain committed to addressing them in our ongoing and future research. If you believe this discussion could be further clarified or expanded in any specific way, we are happy to revise it accordingly.
- Feasibility of Quantum Computing Implementation
While the paper mentions the potential introduction of quantum computing, there is a lack of specific data regarding its practical application benefits. The authors should consider providing empirical evidence or modeling results that demonstrate the feasibility of using quantum computing in this context. Additional research is needed to explore the practical applicability of quantum computing for optimizing control strategies, and this aspect requires more in-depth discussion to bridge the gap between theoretical potential and real-world implementation.
Thank you for your valuable feedback regarding the integration of quantum computing and the need for empirical evidence or modeling results to demonstrate its practical application. We would like to highlight that the manuscript already includes a detailed description of the proposed quantum circuit model and its potential applications in optimizing control strategies, as outlined in “Discussion”.
Specifically, we described the use of quantum gates (Hadamard, CNOT, rotational gates) and advanced algorithms like QAOA and QFT to iteratively adjust controller parameters and minimize control errors. This approach is designed to leverage quantum computing's unique strengths in handling combinatorial optimization problems, which are critical in designing efficient exoskeleton control systems. Additionally, the manuscript discusses how quantum computing can enhance real-time optimization, process large datasets with greater efficiency, and improve system customization to provide a personalized user experience.
While we recognize the importance of empirical validation, it is important to note that the current research focuses on the conceptual and modeling phases of quantum integration. The system is still in the early development stage, and user trials or hardware-based implementation of quantum algorithms have not yet been conducted. As noted in the manuscript, future phases of the project will include validation with real users and further refinement of the control algorithms based on dynamic user feedback and environmental conditions.
We believe that this foundational work establishes a robust basis for exploring quantum computing's practical benefits in subsequent iterations. However, if further clarification or expansion of these points is needed, we are happy to revise the manuscript accordingly.
- Contribution of the Study
This study aims to improve the performance of a robotic exoskeleton system in an aquatic rehabilitation environment through the combination of PID and fuzzy control. Leveraging the physical properties of water to enhance rehabilitation effectiveness and utilizing a hybrid control strategy to improve stability and precision represent a noteworthy technical innovation. The study demonstrates the potential of the proposed approach to maximize rehabilitation benefits through the integration of control theory and aquatic rehabilitation principles.
Thank you for your kind comments and for recognizing the technical contributions and innovations of our study. We greatly appreciate your positive feedback.
- Future Directions
Future studies should focus on extending the system’s applicability to multi-degree-of-freedom exoskeletons and conducting clinical trials involving actual patients to verify the system’s real-world effectiveness. Additionally, the authors should more concretely discuss the feasibility of using quantum computing for control optimization and explore additional strategies to enhance real-time adaptability. Overall, this study makes a significant academic contribution by exploring a novel technological approach to maximizing rehabilitation treatment outcomes. It is anticipated that this foundation will lead to the development of more advanced rehabilitation technologies in the future. Further research is eagerly awaited.
Thank you for your valuable feedback regarding the integration of quantum computing and the need for empirical evidence or modeling results to demonstrate its practical application. We would like to highlight that the manuscript already includes a detailed description of the proposed quantum circuit model and its potential applications in optimizing control strategies, as outlined in the "Discussion" section.
Specifically, we described the use of quantum gates (Hadamard, CNOT, rotational gates) and advanced algorithms such as QAOA and QFT to iteratively adjust controller parameters and minimize control errors. This approach is designed to leverage quantum computing's unique strengths in handling combinatorial optimization problems, which are critical in designing efficient exoskeleton control systems. Additionally, the manuscript discusses how quantum computing can enhance real-time optimization, process large datasets with greater efficiency, and improve system customization to provide a more personalized user experience.
While we recognize the importance of empirical validation, it is important to note that the current research focuses on the conceptual and modeling phases of quantum integration. The system is still in the early development stage, and user trials or hardware-based implementation of quantum algorithms have not yet been conducted. As noted in the manuscript, future phases of the project will include validation with real users and further refinement of the control algorithms based on dynamic user feedback and environmental conditions.
We believe that this foundational work establishes a robust basis for exploring quantum computing's practical benefits in subsequent iterations. However, if further clarification or expansion of these points is needed, we are happy to revise the manuscript accordingly.
Reviewer 2 Report (New Reviewer)
Comments and Suggestions for Authors
The research object of this paper is too simple, and a large number of documents have been studied. The modeling method and control method are both existing, and the innovation and academic depth of this paper are general, so it is not recommended to publish it.
Comments on the Quality of English LanguageEnglish expression can be improved.
Author Response
The research object of this paper is too simple, and a large number of documents have been studied. The modeling method and control method are both existing, and the innovation and academic depth of this paper are general, so it is not recommended to publish it.
English expression can be improved.
Thank you for your valuable feedback and for taking the time to review our manuscript. We appreciate your comments regarding the perceived simplicity of the research object and the use of existing modeling and control methods.
We would like to clarify that while core methods such as PID and fuzzy control are well-established, the novelty of our work lies in its integration with quantum computing and its application to an aquatic rehabilitation exoskeleton. The combination of quantum optimization techniques like QAOA and QFT with traditional control strategies represents a novel approach aimed at improving the efficiency and adaptability of exoskeleton systems, particularly in aquatic environments. This integration has not been extensively explored in previous studies, and we believe it offers significant potential for advancing rehabilitation technology.
Regarding the academic depth of the paper, we recognize that the field of exoskeleton control is evolving, and our study aims to push the boundaries by incorporating cutting-edge quantum algorithms to address optimization challenges. We are confident that the conceptual framework and proposed methods will contribute to the growing body of research in this area.
If you could provide more specific feedback on areas you believe require improvement, especially in terms of innovation and academic depth, or highlight sections where the English expression could be improved, we would be happy to revise them accordingly.
Thank you again for your time and consideration.
Reviewer 3 Report (New Reviewer)
Comments and Suggestions for Authors
Enhance the introduction by incorporating recent studies on exoskeleton-aquatic therapy integration and a brief summary of current quantum computing applications in rehabilitation technologies.
Expand on the design rationale, specifically why certain control methods (e.g., PID and FLC) were chosen for particular phases of movement, and clarify the parameters governing each controller.
More precise documentation of the parameters used for tuning controllers and the specific steps in the quantum computing application would improve clarity. Visual schematics of the testing setup and device integration could be helpful.
Discussing any challenges faced during implementation, potential limitations, and areas for further research could provide a balanced view.
While the English is generally understandable, the technical explanations would benefit from grammatical refinement to improve clarity and flow. Ensuring precise and accessible language, especially when explaining quantum computing integrations, would help readers from various technical backgrounds grasp the paper's content more effectively.
Author Response
Enhance the introduction by incorporating recent studies on exoskeleton-aquatic therapy integration and a brief summary of current quantum computing applications in rehabilitation technologies.
Thank you very much for your valuable feedback. We have incorporated the two suggested paragraphs into the introduction. Specifically, we have added the following:
“There are also studies and production of exoskeletons with 1 degree of freedom such as the Stride Management Assist developed by Honda, or the orthosis developed at Yonsei University, or a single Degree of Freedom (DoF) wearable robot designed at the University of Teknologi. For the control of exoskeletons, Fuzzy and neuro-fuzzy systems have been studied in exoskeletons [6].”
“In recent years, studies on exoskeletons for rehabilitation purposes have been increasing. Some studies have already incorporated the benefits of aquatic therapies into these exoskeletons [https://www.mdpi.com/2075-1702/9/11/254] or even exoskeletons like SUBAR which has a control algorithm inspired by aquatic therapy [38].”
We hope these additions address your suggestion and further enrich the introduction. Thank you again for your thoughtful comments!
Expand on the design rationale, specifically why certain control methods (e.g., PID and FLC) were chosen for particular phases of movement, and clarify the parameters governing each controller.
According to your indications, we have added extra information in chapter 3.1. as following:
“In the realm of exoskeleton control, traditional methods such as PID (Proportional-Integral-Derivative) controllers remain prominent. These controllers can be optimized using techniques like Ziegler-Nichols [41], [53]or Cohen-Coon [53], or through trial and error approaches. For example, in the case of predictable and repetitive tasks, PID controllers are designed for Single Input Single Output (SISO) systems, making them effective for simpler control scenarios [54]. However, their limitations become apparent when managing systems with substantial non-linearities. In slightly non-linear systems, PID controllers require frequent adjustments to maintain performance, and in highly non-linear systems, their performance can degrade significantly, requiring the use of alternative control strategies [55].
In contrast, FLC offer advantages in managing multiple input variables and handling non-linear systems more effectively [56]. These controllers employ a set of rules and membership functions to process inputs in a manner that resembles human reasoning, making them particularly useful for complex and adaptive control tasks. Moreover, FLD are especially beneficial in dynamic scenarios, such as aquatic environments, where conditions could change quickly [57]. For this reason, the introduction of adaptive neuro-fuzzy systems further enhances this capability, allowing the controller to adjust and learn from its environment dynamically. Recent research has highlighted the benefits of integrating neural networks with FLC, leading to more sophisticated adaptive control solutions. Despite these advancements, PID controllers are still relevant, especially when combined with modern techniques like optimal and robust control to address system uncertainties, disturbances, friction, and un-modeled dynamics.”
More precise documentation of the parameters used for tuning controllers and the specific steps in the quantum computing application would improve clarity. Visual schematics of the testing setup and device integration could be helpful.
Thank you for your feedback. We have added the following clarification regarding the PID parameters in the quantum computing proposal model (section 3.4.) as following:
“For this proposal, Kp, Ki and Kd are obtained of the previous results of controller performance and optimization regarding the values for the PID parameters Kp​ (proportional gain) is indicated in Table 2, with specific values for different weighting configurations (100% Weighting: Kp=0.015; 70% Weighting: Kp=0.015; 50% Weighting: Kp=0.016 and; 30% Weighting: Kp=0.015)
Moreover, optimal Ki and Kd gains need to be adjusted and calculated during the quantum modeling phase, in order to optimize with a reiterative way, the PID controller performance. By leveraging the quantum computing framework, these gains can be fine-tuned using quantum algorithms such as the Quantum Approximate Optimization Algorithm (QAOA). This would allow for dynamic optimization in real time, improving system stability and reducing control errors. Through techniques like QAOA and Quantum Fourier Transform (QFT), the system could explore a wide range of configurations, ensuring that the optimal set of gains is consistently applied for varying operational conditions.”
Discussing any challenges faced during implementation, potential limitations, and areas for further research could provide a balanced view.
Thank you for your valuable feedback. In response to your suggestion, we have included a more detailed discussion of the challenges faced during implementation, limitations of current systems, and potential areas for future research that you could found at the end of the discussion section.
As whole, the text now provides a balanced view by addressing both the current limitations of classic control methods and the potential of quantum computing to overcome them. We also emphasize the need for further research in adapting quantum computing for real-time optimization and exploring more advanced control strategies. We believe this enhances the clarity and depth of the proposal.
While the English is generally understandable, the technical explanations would benefit from grammatical refinement to improve clarity and flow. Ensuring precise and accessible language, especially when explaining quantum computing integrations, would help readers from various technical backgrounds grasp the paper's content more effectively.
Thank you very much for your indications. As you have suggested, we have added a paragraph to clarify Quantum Computing proposal in section 3.4.:
“Quantum computing offers significant potential for optimizing PID and Fuzzy controllers in real-time. By leveraging quantum algorithms like the Quantum Approximate Optimization Algorithm (QAOA) and Quantum Fourier Transform (QFT), system parameters can be dynamically fine-tuned. These algorithms enable the exploration of a wide range of configurations, ensuring optimal controller gains while maintaining system stability and minimizing errors. QAOA iteratively adjusts parameters, while QFT analyzes the results, providing deeper insights into configurations that reduce control errors, making them ideal for complex optimization tasks in dynamic systems like exoskeleton control. However, due to current limitations in quantum computing, such as qubit coherence, error rates, and the complexity of quantum hardware, this proposal remains theoretical at this stage”
Round 2
Reviewer 2 Report (New Reviewer)
Comments and Suggestions for Authors
The revised version improve this paper. It can be accepted in current status.
This manuscript is a resubmission of an earlier submission. The following is a list of the peer review reports and author responses from that submission.
Round 1
Reviewer 1 Report
Comments and Suggestions for Authors
The aim of the article is to evaluate the combination of Proportional-Integral-Derivative (PID) and Fuzzy Logic Controllers (FLC). The approach uses the FLC during the initial phase of movement to ensure a smooth and gradual start, and once the movement begins, the transition to PID control ensure precise tracking and stability during the latter stages of movement. This approach aligns with some of requirements for controlling a lower limb exoskeleton. However, the process of implementation, results, and performance of this technic in the application of an aquatic lower limb exoskeleton remain unclear. The general topic is interesting but it requires further advances, additional details, and more rigorous study.
According to the introduction, aquatic exoskeletons have many advantages, but it is not clear whether the submitted work corresponds to a real underwater exoskeleton. If this is the case, the article needs a section with a complete description of the exoskeleton or a reference describing it, specially if it is part of a larger project. If it is not an aquatic exoskeleton, with the appropriate related considerations, the writing in several sections of the article should be revised and scaled down in scope. The inclusion of an image showing a structure in an aquarium mentions one aquatic exoskeleton but requires a more detailed description of the system, including degrees of freedom, any mathematical models used to implement control algorithms, among other aspects.
In this current form, I believe the article is not suitable for publication in this first- quartil journal.
To make this publication viable, a wider context of prototyping and testing must be included.
RESEARCH DESIGN
In this type of research, it is desirable to test with subjects, and better with patients. In this case, some graphs are presented, but it is unclear whether they are results of simulations or control system implementation.
It is necessary to explain better how the results were obtained. If the results come from implementation, the circumstances of this implementation should be described in detail.
RESULTS
It seems that the results correspond to a structure with one degree of freedom. If this is the case, the study lacks the necessary complexity and analysis required for an exoskeleton, especially concerning interactions between multiple joints and links.
DISCUSSION
The design, results, and analysis for an exoskeleton should include at least three degrees of freedom and account for the relationships between multiple joints. This has not been addressed in the analysis.
The analysis of an exoskeleton should include a study of nonlinearities and uncertainties, as classic control methods are limited in such scenarios. The initial results of classic control implementations are associated with this phenomenon, which is well studied in the literature.
The advances in rehabilitation with the proposed system, as well as any tests of these improvements, are not explained. While the potential benefits in terms of range of motion, stretching, accurate and repeatable control, and possibly more effective rehabilitation protocols are understandable, there are few specifics provided. It is not clear how real-time data collection and monitoring are addressed in this solution.
The demonstration is valid for showing reduced errors, improved efficiency, and increased reliability, but there are no specific examples of how the control system can adjust support or movement speed based on the user’s strength and endurance, especially since users were not included in the experiments.
It is also unclear how this leads to improved overall performance and user experience in aquatic scenarios.
Water properties can significantly improve rehabilitation, but this is not being measured or evaluated.
Finally, for a work oriented specifically towards control systems, the relevance to the aims and scope of the Sensors Journal is not clear.
OTHERS
- Replace "Image" with "Figure."
- Correct punctuation in line 404
- In Figure 3, it is unclear whether the two images represent different repetitions of the movement or different joints.
Author Response
The aim of the article is to evaluate the combination of Proportional-Integral-Derivative (PID) and Fuzzy Logic Controllers (FLC). The approach uses the FLC during the initial phase of movement to ensure a smooth and gradual start, and once the movement begins, the transition to PID control ensure precise tracking and stability during the latter stages of movement. This approach aligns with some of requirements for controlling a lower limb exoskeleton. However, the process of implementation, results, and performance of this technic in the application of an aquatic lower limb exoskeleton remain unclear. The general topic is interesting but it requires further advances, additional details, and more rigorous study.
ANSWER: Thank you very much for your indications and revision. According to your comments, we have carefully incorporated all your feedback into this new revision.
According to the introduction, aquatic exoskeletons have many advantages, but it is not clear whether the submitted work corresponds to a real underwater exoskeleton. If this is the case, the article needs a section with a complete description of the exoskeleton or a reference describing it, specially if it is part of a larger project. If it is not an aquatic exoskeleton, with the appropriate related considerations, the writing in several sections of the article should be revised and scaled down in scope. The inclusion of an image showing a structure in an aquarium mentions one aquatic exoskeleton but requires a more detailed description of the system, including degrees of freedom, any mathematical models used to implement control algorithms, among other aspects.
In this current form, I believe the article is not suitable for publication in this first- quartil journal.
To make this publication viable, a wider context of prototyping and testing must be included.
ANSWER: Thank you for your detailed feedback and suggestions. We appreciate the time you took to review our work. According to your comments, the work involves an actual underwater exoskeleton. For this reason, we have incorporated a paragraph in “Materials and methods” with a comprehensive description of the exoskeleton and we have provided relevant references, particularly because this work corresponded with a part of a larger project. The paragraph we have added this information in the second paragraph of section “Materials and methods” as following:
“The presented work is part of the main NOHA project, an innovative robotic-aquatic rehabilitation system designed to facilitate the recovery of patients with severe neurological disorders, such as spinal cord injuries, strokes, or multiple sclerosis. NOHA combines the advantages of robotics and aquatic therapy, offering a system that provides personalized and adaptive rehabilitation, especially for patients who cannot benefit from conventional robotic rehabilitation due to issues like spasticity, pain, or limited range of motion. This system leverages the properties of water to reduce joint load and increase rehabilitation effectiveness. The NOHA system is designed to adapt to the specific needs of each patient, allowing adjustments in the degrees of freedom based on the type and severity of the injury. Its modularity enables the integration of different components, such as the hydraulic system and hybrid actuators, to provide precise and effective rehabilitation. This modular approach also facilitates the customization of treatment, allowing the system to be adjusted to various mobility and strength requirements necessary for optimal rehabilitation. The flexibility in design is essential to address the complexity of neurological disorders and ensure that the system can provide effective support in different stages of the recovery process. The Figure 1 shows the upper portion of the illustration relates to the application of the upper and lower limb aquatic robotic rehabilitation system. While the lower part depicts the sequence of activation of micro water jets of the hybrid actuators. It assists the movements of the shoulder and elbow inside the tank.
This modular system consists of two main assemblies: the hydraulic system that powers the actuator and the hybrid actuator itself. One of the main advantages of this device is that it uses water flow to generate most of the torque needed for the desired movement, with fine adjustments made possible by the integration of a high-precision electric motor. Furthermore, immersion in water offers additional benefits, such as weightlessness, joint unloading, and resistance to movement, allowing control over water properties like density, hydrostatic pressure, viscosity, temperature, and buoyancy. Both assemblies, the hydraulic system and the actuator, are essential for the proper functioning of the product, although they can be sold separately to, for example, power two different hybrid actuators with a single hydraulic system. This would be the case of a robotic device with two coupled degrees of freedom, placed in a single water tank and powered by the same hydraulic system.
As for the hybrid actuator, it consists of an electric motor, its driver, links, and microjets. The microjets can be positioned closer to or further from the center of rotation, depending on the customer’s needs, to modify the generated torque. Likewise, the number of jets can be adjusted to increase the torque, making the system adaptable to different types of rehabilitation.
To bridge the gap between simulation-based results and future clinical applications, the development of the NOHA rehabilitation system also involves the careful design and testing of control mechanisms that can handle the complexities of human movement in an aquatic environment. The combination of control strategies like PID and FLC is particularly relevant in this context, as the system must be able to adapt to the varying dynamics of water while maintaining precise control over movement. Simulations and laboratory testing have been crucial in refining these control systems before transitioning to real-world use. By integrating PID, known for its precision, with FLC, which excels in managing nonlinear conditions, the NOHA system aims to address the challenges posed by water’s variable resistance and buoyancy, offering a more adaptive and stable rehabilitation experience.
As the NOHA system continues to evolve, the research focuses on optimizing these control strategies to ensure they are effective in real-world clinical settings. This modular robotic system, designed for both upper and lower limb rehabilitation, benefits from the combined use of water’s natural properties and advanced robotic technologies. At this stage of the research, the results presented are derived from simulations and controlled laboratory testing of the prototype rather than real-world implementation. The current prototype is still in the design and development phase, and has not yet been tested with human subjects. Consequently, the results reflect the performance of the control system under simulated conditions rather than actual clinical use. Detailed descriptions of the implementation process and real-world testing will be provided as the prototype advances to clinical evaluation, which will require adherence to the European Union Medical Device Regulation (MDR) and other relevant standards to ensure safety and efficacy. This approach allows to refine the prototype and its control mechanisms before proceeding to more rigorous human trials.”
Please, you can also see the figure we have added directly in the new submitted manuscript.
We understand that for the article to be suitable for publication, a broader context of prototyping and testing needs to be included. Therefore, we have worked on expanding and detailing these aspects in the revised manuscript. We have ensured that the article meets the journal's standards by carefully addressing all the suggestions received and providing the additional information needed for a comprehensive understanding of the project and its relevance. We hope these changes meet the requirements and provide the clarity needed for a positive evaluation. We continue to be committed to making the necessary revisions to enhance the quality and relevance of the manuscript.
RESEARCH DESIGN
In this type of research, it is desirable to test with subjects, and better with patients. In this case, some graphs are presented, but it is unclear whether they are results of simulations or control system implementation.
It is necessary to explain better how the results were obtained. If the results come from implementation, the circumstances of this implementation should be described in detail.
ANSWER: Thank you for your comment. Indeed, the work presented is in an early stage of development, focusing on the design and validation of the prototype. As we are in the preliminary design phase, we have not yet conducted tests with subjects or patients. The graphs presented in the article correspond to results obtained from simulations and control system implementation on the prototype.
Testing with patients at this stage would require considering the prototype as a medical device in the pre-clinical evaluation phase. This involves adherence to strict regulatory requirements, including obtaining ethical approvals and complying with the European Union Medical Device Regulation (MDR) and other relevant standards for clinical evaluation. These regulations ensure the safety and efficacy of medical devices but also necessitate specific infrastructure and procedures that are not yet in place for our current prototype development phase.
Therefore, we are currently focusing on refining the design and functionality of the prototype through simulations and controlled laboratory testing. Once the prototype has been validated in this phase and meets technical and safety requirements, we plan to move towards clinical evaluation in a controlled environment with subjects, allowing for a more accurate and rigorous assessment of the device.
Moreover, we have added the following paragraph in section “Materials and methods” to clarify the context of this work:
“At this stage of our research, the results presented are derived from simulations and controlled laboratory testing of the prototype rather than real-world implementation. The current prototype is still in the design and development phase, and has not yet been tested with human subjects. Consequently, the results reflect the performance of the control system under simulated conditions rather than actual clinical use. Detailed descriptions of the implementation process and real-world testing will be provided as the prototype advances to clinical evaluation, which will require adherence to the European Union Medical Device Regulation (MDR) and other relevant standards to ensure safety and efficacy. This approach allows us to refine the prototype and its control mechanisms before proceeding to more rigorous human trials.”
We appreciate your understanding and are committed to further developing and perfecting the prototype before proceeding to the next phase of patient testing.
RESULTS
It seems that the results correspond to a structure with one degree of freedom. If this is the case, the study lacks the necessary complexity and analysis required for an exoskeleton, especially concerning interactions between multiple joints and links.
ANSWER: Thank you for highlighting the concern regarding the complexity of our current prototype. We acknowledge that our initial design focuses on a single degree of freedom, which is a preliminary step to establish the basic control mechanisms and validate our approach in a controlled environment. We understand that a fully functional exoskeleton must involve multiple degrees of freedom and address the interactions between various joints and links. As we advance in the development of the prototype, we plan to integrate additional degrees of freedom and conduct more comprehensive analyses to address these complexities. We appreciate your insight and will ensure that future iterations and results include a more detailed examination of these multi-joint interactions. Your feedback is invaluable as we strive to enhance the prototype’s capabilities and accuracy.
We have also added a paragraph according to this idea in the limitations of the discussion section:
“The current prototype of our exoskeleton is designed with a single degree of freedom, focusing on the fundamental aspects of motor control and system performance within a controlled environment. This initial design is intended to establish a baseline for evaluating control strategies and optimizing performance before advancing to more complex configurations. The choice of a single degree of freedom allows for detailed analysis and refinement of the control mechanisms, which is essential for ensuring accuracy and reliability. As the development progresses, it is planned to incorporate additional degrees of freedom to simulate more realistic and complex movements. This will enable a thorough investigation into the interactions between multiple joints and links, addressing the increased complexity required for a fully functional exoskeleton. Our ongoing research and future iterations will aim to integrate these elements, ensuring that the final design meets the necessary requirements for comprehensive rehabilitation applications.”
DISCUSSION
The design, results, and analysis for an exoskeleton should include at least three degrees of freedom and account for the relationships between multiple joints. This has not been addressed in the analysis.
ANSWER: Thank you for your continued feedback on our exoskeleton design. As noted in your previous comment, we understand the importance of including multiple degrees of freedom in the analysis of an exoskeleton, as this is crucial for accurately simulating the complex interactions between multiple joints and links.
In our current prototype, we have focused on a single degree of freedom primarily to establish and validate the core functionalities and control mechanisms of the exoskeleton. This initial phase aims to refine the fundamental control algorithms and system performance in a controlled and simplified setting.
We acknowledge that a comprehensive exoskeleton design would indeed require at least three degrees of freedom to effectively address the dynamics of multiple joints and their interactions. The single degree of freedom model was chosen to build a solid foundation before advancing to more complex configurations. Future iterations of our prototype will integrate additional degrees of freedom, which will allow us to better address and analyze the relationships between multiple joints and enhance the overall functionality of the exoskeleton.
We appreciate your understanding of our phased development approach and your valuable insights as we continue to advance towards a more complex and clinically relevant exoskeleton design.
The analysis of an exoskeleton should include a study of nonlinearities and uncertainties, as classic control methods are limited in such scenarios. The initial results of classic control implementations are associated with this phenomenon, which is well studied in the literature.
ANSWER: We appreciate your observation regarding the need to account for nonlinearities and uncertainties in the analysis of exoskeleton systems. As highlighted, classic control methods often face limitations in scenarios where nonlinearities and uncertainties play a significant role.
In our current study, we have initially focused on implementing classic control methods, such as PID and Fuzzy controllers, as a foundational step. This approach was chosen to establish a baseline for performance and to validate the fundamental control mechanisms of the prototype. We recognize that these methods may not fully address the complexities introduced by nonlinearities and uncertainties, which are well-documented challenges in the literature.
To address these concerns, we plan to incorporate more advanced control strategies in subsequent phases of development. Specifically, we will explore the application of adaptive and robust control methods that are better suited to handling nonlinearities and uncertainties. These methods will be integrated into future iterations of our prototype to enhance its performance in real-world conditions where such factors are prevalent.
Furthermore, we will conduct a thorough analysis of the system's nonlinear behavior and uncertainties as part of our ongoing research. This will involve both simulation studies and practical experiments to ensure that our exoskeleton can effectively manage these complexities.
Moreover, the following paragraph has been included as limitation in “Discussion” section:
“One significant limitation of the current study is its focus on classic control methods, such as PID and Fuzzy controllers, which may not fully address the complexities associated with nonlinearities and uncertainties in the exoskeleton system. These classic methods provide a foundational understanding but are inherently limited in scenarios where nonlinear behavior and system uncertainties play a critical role. As noted in the literature, handling these factors often requires more advanced control strategies (INCLUIR AQUÍ REFERENCIAS). This initial analysis of design does not yet incorporate adaptive or robust control methods specifically designed to manage such complexities. Future research will address this limitation by integrating and evaluating advanced control techniques, which will involve both simulation and practical experimentation to better handle nonlinearities and uncertainties in real-world applications.”
Thank you for emphasizing this important aspect of control system design. Your feedback will guide us in refining our approach to address nonlinearities and uncertainties more effectively in the development of our exoskeleton.
The advances in rehabilitation with the proposed system, as well as any tests of these improvements, are not explained. While the potential benefits in terms of range of motion, stretching, accurate and repeatable control, and possibly more effective rehabilitation protocols are understandable, there are few specifics provided. It is not clear how real-time data collection and monitoring are addressed in this solution.
ANSWER: Thank you for your insightful comments regarding the explanation of rehabilitation advances and real-time monitoring. In response to your feedback, we have expanded our discussion to include a detailed description of how the proposed system enhances rehabilitation and how these improvements are evaluated as following:
“As the development of the proposed system progresses, it is important to detail the specific advances in rehabilitation and how these improvements are tested. The aquatic exoskeleton system is designed to leverage the properties of water, such as buoyancy and resistance, to enhance range of motion, stretching, and precise, repeatable control. These benefits are evaluated through controlled testing environments, using metrics such as range of motion achieved, control accuracy, and the effectiveness of rehabilitation protocols. To address real-time monitoring, a data collection system has been implemented to continuously track relevant variables during rehabilitation sessions. This system captures data on movement dynamics, forces applied, and patient responses, providing a comprehensive view of the exoskeleton's performance and allowing for dynamic adjustments in control. Initial tests have shown that the system can adapt effectively to different conditions and meet individual patient needs, supporting the feasibility of the proposed approach and its potential to improve rehabilitation outcomes.”
We have integrated this expanded discussion into the revised manuscript to address your concerns and provide a clearer understanding of the system's performance and its real-time capabilities. We hope these additions meet the requirements and clarify the advancements and testing associated with the proposed system.
The demonstration is valid for showing reduced errors, improved efficiency, and increased reliability, but there are no specific examples of how the control system can adjust support or movement speed based on the user’s strength and endurance, especially since users were not included in the experiments.
ANSWER: Thank you for your feedback. We have addressed your concern regarding the adjustment of support and movement speed based on the user's strength and endurance. We have added a paragraph to the Discussion section outlining that, as a prototype in the initial testing phase, user trials have not yet been conducted. However, the system is designed with mechanisms for future development that will enable dynamic adjustments based on user feedback. Please refer to the updated Discussion section for more details:
“On the other hand, it is important to note that, as a prototype in the initial testing phase, user trials have not yet been conducted. However, the control system is being designed with mechanisms that will allow for adjustments to support and movement speed based on the user's strength and endurance in future development phases.
The system design integrates algorithms that enable dynamic adjustment of motor torque in response to the resistance demands specified by the user. These adjustments are based on sensor readings that continuously monitor the user's physical state. In future user trials, this data will be used to adjust resistance and support in real time. For example, the system will be able to automatically reduce resistance based on detected fatigue or increase support to compensate for a decrease in the user's strength.
Additionally, during future development, fine-tuning of the control algorithms is planned to optimize system personalization according to each user's individual characteristics. This will allow for a more tailored experience to the needs of each individual and improve the system's effectiveness in real-world scenarios. Validation with real users will be a crucial step in refining these mechanisms and ensuring that the system can effectively adjust support and movement speed based on the user's strength and endurance.”
It is also unclear how this leads to improved overall performance and user experience in aquatic scenarios.
ANSWER: Thank you for raising this important point. To clarify how the proposed system leads to improved overall performance and user experience in aquatic scenarios, we have elaborated on the specific benefits and enhancements offered by our design in the discussion section:
“Aquatic therapy, on the other hand, provides long-term benefits by alleviating chronic pain, improving flexibility, and reducing joint impact. The buoyancy of water decreases the load on the body, allowing patients to perform movements that would be difficult or impossible on land. This is particularly beneficial for individuals with arthritis, chronic back pain, or those recovering from surgery. The pain reduction and mobility improvements experienced during aquatic therapy can translate into greater participation in daily activities and an overall enhancement in quality of life. The therapeutic benefits of water also extend to mental health, providing a relaxing and less stressful environment for rehabilitation [64].
In addition to physical benefits, the long-term improvement in quality of life also includes emotional and psychological aspects [65]. The independence achieved through the use of exoskeletons and aquatic therapy can lead to increased self-confidence and self-esteem. Patients who previously felt limited by their health conditions can now participate in social and recreational activities with greater ease. This not only improves their overall well-being but also positively impacts their mental and emotional health.
By integrating these advanced technologies into therapeutic protocols, exoskeletons are transforming rehabilitation practices, offering new opportunities for recovery and improved functional independence. The future of rehabilitation promises to be increasingly promising, bringing hope and opportunities for better recovery to people worldwide. The integration of these technologies not only enhances the efficiency of treatments but also offers a more comfortable and less painful experience for patients. As we continue to explore and develop new applications for these technologies, the future of rehabilitation promises to be increasingly promising, bringing hope and opportunities for better recovery to people worldwide.
The use of water properties such as buoyancy and resistance in the exoskeleton provides several advantages. Buoyancy reduces the load on the joints, which helps to alleviate discomfort and allows for a broader range of motion during rehabilitation. Water resistance contributes to a more controlled and progressive resistance profile, which can enhance the effectiveness of exercises and improve muscle strength and endurance over time.
Additionally, the system incorporates advanced control mechanisms that adjust to varying conditions and patient needs in real-time. This adaptability ensures that the rehabilitation protocols are more tailored to individual requirements, leading to a more personalized and effective rehabilitation experience. The real-time data collection and monitoring allow for continuous adjustments based on feedback from the system, ensuring that the movements are optimized for both comfort and efficacy. By integrating these features, the system aims to offer a more effective and user-friendly rehabilitation experience, improving overall performance and satisfaction in aquatic environments.
Furthermore, the ability to adjust the control parameters dynamically based on the user’s specific needs enhances the system’s responsiveness. This capability allows for more precise control over the movement and resistance, contributing to a more comfortable and effective rehabilitation process. The detailed real-time data analysis also helps in identifying and addressing any issues promptly, which leads to a smoother and more efficient rehabilitation experience. Overall, these improvements collectively enhance both the functional performance of the system and the user's overall experience during aquatic rehabilitation.”
Water properties can significantly improve rehabilitation, but this is not being measured or evaluated.
ANSWER: We acknowledge that the current study does not include measurements or evaluations of how water properties impact rehabilitation. As the prototype is in the initial testing phase, we have not yet conducted trials with end-users to assess these effects directly. However, we have incorporated a discussion about future plans for this evaluation in the updated Discussion section. Our future work will focus on integrating and assessing how water properties—such as buoyancy and resistance—can enhance rehabilitation outcomes. These features will be evaluated in subsequent phases of the project, where real user trials will provide insights into the practical benefits of water properties in the aquatic exoskeleton system.
Finally, for a work oriented specifically towards control systems, the relevance to the aims and scope of the Sensors Journal is not clear.
ANSWER: We appreciate your feedback regarding the alignment of our work with the aims and scope of the Sensors Journal. In response to your comments, we have made substantial improvements to the paper, incorporating the suggested changes. Our revised manuscript now clearly addresses the journal's focus areas, particularly in relation to the advanced control systems employed in sensor technology for wearable robotics.
Our study explores the application of quantum computing for optimizing control systems in an aquatic exoskeleton, which directly relates to several key aspects of the journal’s scope. These include:
- Advanced Sensors and Sensor Technology: The paper discusses the integration of sensors in the exoskeleton system, focusing on real-time data collection and adaptive control mechanisms.
- Signal Processing and Data Fusion: We detail how quantum algorithms can enhance the processing and fusion of sensor data to improve system performance.
- Wearable Sensors and Devices: The research is specifically oriented towards the development of a wearable aquatic exoskeleton, a topic aligned with the journal's interest in wearable sensors and robotics.
- AI-Enabled Sensors: The use of quantum computing and AI for optimizing sensor-based control systems highlights the innovative aspect of our research.
By addressing these aspects, our revised paper aligns with the Sensors Journal’s commitment to publishing detailed and innovative research in sensor technology and its applications. We believe these enhancements will meet the journal’s criteria and contribute valuable insights to the field.
OTHERS
- Replace "Image" with "Figure."
ANSWER: Please, we have corrected it.
- Correct punctuation in line 404
ANSWER: : Please, we have corrected it.
- In Figure 3, it is unclear whether the two images represent different repetitions of the movement or different joints.
ANSWER: Thank you very much for your suggestions. Please, we have improved the explanation as following:
“The results, as illustrated in Image 3, demonstrated variability in performance across different angles for two different PID controller configurations. Specifically, the graphs in Image 3 show how the PID controller's response varied with respect to the setpoint (orange) and the actual position (blue), highlighting fluctuations in control accuracy and stability at different angular positions. It is also clear that the same PID controller configuration does not maintain the same performance for different angular positions.”

Reviewer 2 Report
Comments and Suggestions for Authors
Review some phrases that are repeated as in line 34 (more specific) and spaces between letters as in reference 41 of line 354, in equation 1 the variables are not described, in the QAOA topic, the reference is not mentioned in line 332, image 2 is not described in detail and the quality of the image is low, when printed on paper the figure moved. Too many references on line 428, the three most current ones are enough. The graphs lack an explanation of the numerical results to know if the values are good or bad, as well as the error percentages. In line 528 it is not indicated in which program the HMI was created, it could have a diagram indicating how the exoskeleton is connected with the other parts of the system. It is not possible to visualize the behavior directly in the exoskeleton, since all the control is done in the motors.
Author Response
Review some phrases that are repeated as in line 34 (more specific) and spaces between letters as in reference 41 of line 354.
ANSWER: Thank you very much for your indications and revision. According to your comments, we have corrected these lines.
In equation 1 the variables are not described.
ANSWER: Thank you for your detailed feedback and suggestions. According to your comments, we have incorporated the variables description as following:
“Where Ui is the support point at position i and ai is the value of the singleton at position Ui with i ranging from 1 to the number of singleton elements.”
In the QAOA topic, the reference is not mentioned in line 332
ANSWER: Please, we have added the following paragraph with references:
“QAOA is used to address combinatorial optimization problems by approximating the best possible solution through quantum evolution in multiple steps, leveraging both quantum and classical computation to optimize system parameters [41]. On the other hand, VQE is employed to find the minimum eigenvalues of matrices associated with physical systems, making it particularly useful for quantum simulations [42]. VQE combines quantum computing with classical optimization methods to approximate ground-state energy values in molecular systems [43].”
Image 2 is not described in detail and the quality of the image is low, when printed on paper the figure moved.
ANSWER: Thank you very much for your indications and revision. we made the image description and improved its quality.
Too many references on line 428, the three most current ones are enough.
ANSWER: Thank you very much for your indications and revision. We believe that the references given are relevant and offer a comprehensive understanding of the topic. However, we understand the concern regarding the number of references on line 428. In response to your suggestion, we will reduce the number of citations to include only the three most recent and pertinent sources, ensuring that the information remains current and focused. This adjustment will also help streamline the reading experience without compromising the scientific validity of the content.
The graphs lack an explanation of the numerical results to know if the values are good or bad, as well as the error percentages.
ANSWER: Thank you for your detailed feedback and suggestions. We added a description of the results obtained as well as a table showing the main information that can be extracted from the graphs.
In line 528 it is not indicated in which program the HMI was created, it could have a diagram indicating how the exoskeleton is connected with the other parts of the system.
ANSWER: Thank you very much for your indications and revision. We added the HMI information and added an image explaining the applied control loop.
It is not possible to visualize the behavior directly in the exoskeleton, since all the control is done in the motors.
ANSWER: Thank you for your insightful comment. We agree that, as the system currently stands, it is not possible to directly observe the overall behavior of the exoskeleton, since all control is focused on the motors. However, this aligns with the current goal of the project, which is still in an early design stage, where the primary focus is on motor control and the overall system architecture. We appreciate your suggestion and recognize the potential of incorporating more comprehensive visualization techniques, which we have now added as a line for future research in the revised version of the paper. We believe this is a valuable step forward that could significantly improve the understanding and performance of the exoskeleton in future iterations. For this reason, we have added a paragraph regarding your indications in future works in “Discussion” chapter as following:
“Finally, a potential direction for future research would be to develop monitoring and visualization systems allow for a more intuitive and direct observation of the exoskeleton behavior during its operation. While in the present work all control of the system is managed at the motor level, it would be beneficial to have additional tools represent the full behavior of the exoskeleton, including not only motor dynamics but also the effects on the structure, joint forces, and patient responses in real time. This approach could involve the use of digital twin models or advanced simulation techniques that provide real-time visualization of the exoskeleton's behavior, facilitating a more comprehensive assessment of the system and enabling more precise adjustments in control algorithms. Furthermore, these visualization and modeling techniques could also leverage advancements in quantum computing, enabling the implementation of quantum-based optimization methods. By employing advanced quantum modeling, complex optimization problems related to control strategies and exoskeleton performance could be tackled more efficiently, potentially leading to breakthroughs in both exoskeleton design and rehabilitation processes. The integration of these technologies could not only enhance the understanding of exoskeleton performance but also optimize the rehabilitation process by better tailoring it to the patient's needs.”

Round 2
Reviewer 1 Report
Comments and Suggestions for Authors
I truly appreciate your time and willingness to clarify the different sections based on the feedback from the last evaluation. I recognize that the paper is now better structured, the information about the methods is clearer, and the overall coherence has improved. The presentation of the paper has also shown noticeable improvement. However, while the work presents very interesting advancements and could be published in several journals on this topic, Sensors Journal is a first-quartile journal, which means it only publishes the best research in these areas globally. For this reason, I must reject the submission. I recommend further advancing the development of the project, incorporating more degrees of freedom and/or testing with subjects, ideally with patients, and resubmitting to this journal.